



# Partitioning soil organic carbon into its centennially stable and active fractions with statistical models based on Rock-Eval® thermal analysis (PARTY$_{SOC}$v2.0 and PARTY$_{SOC}$v2.0$_{EU}$)

Lauric Cécillon[1,2], François Baudin[3], Claire Chenu[4], Bent T. Christensen[5], Uwe Franko[6], Sabine Houot[4], Eva Kanari[2,3], Thomas Kätterer[7], Ines Merbach[8], Folkert van Oort[4], Christopher Poeplau[9], Juan Carlos Quezada[10,11,12], Florence Savignac[3], Laure N. Soucémarianadin[13], Pierre Barré[2]

[1]Laboratoire ECODIV, Univ. Normandie, UNIROUEN, INRAE, FR Scale CNRS 3730, Rouen, 76000, France
[2]Laboratoire de Géologie, CNRS, École normale supérieure, PSL University, IPSL, Paris, France
[3]Institut des Sciences de la Terre de Paris, Sorbonne Université, CNRS, Paris, 75005, France
[4]UMR 1402 ECOSYS, INRAE, AgroParisTech, Univ. Paris Saclay, Thiverval-Grignon, 78850, France
[5]Department of Agroecology, Aarhus University, AUFoulum, 8830 Tjele, Denmark
[6]Department of soil system science, Helmholtz Centre for Environmental Research, UFZ, 06120 Halle Germany
[7]Department of Ecology, Swedish University of Agricultural Sciences, 75007 Uppsala, Sweden
[8]Department Community Ecology, Helmholtz Centre for Environmental Research, UFZ, 06246 Bad Lauchstädt, Germany
[9]Thünen Institute of Climate-Smart Agriculture, 38116 Braunschweig, Germany
[10]Laboratory of Ecological Systems ECOS and Laboratory of Plant Ecology Research PERL, School of Architecture, Civil and Environmental Engineering ENAC, École Polytechnique Fédérale de Lausanne EPFL, 1015 Lausanne, Switzerland
[11]Swiss Federal Institute for Forest, Snow and Landscape Research WSL, 1015 Lausanne, Switzerland
[12]Ecosystem Management, Institute of Terrestrial Ecosystems, Department of Environmental Systems Science, ETHZ, 8092 Zürich, Switzerland
[13]ACTA – les instituts techniques agricoles, 75595 Paris, France

*Correspondence to*: Lauric Cécillon (lauric.cecillon@inrae.fr)



**Abstract**

Partitioning soil organic carbon (SOC) into two kinetically different fractions that are centennially stable or active is key information for an improved monitoring of soil health and for a more accurate modelling of the carbon cycle. However, all existing SOC fractionation methods isolate SOC fractions that are mixtures of centennially stable and active SOC. If the stable SOC fraction cannot be isolated, it has specific chemical and thermal characteristics that are quickly (*ca.* 1 h per sample) measureable using Rock-Eval® thermal analysis. An alternative would thus be to (1) train a machine-learning model on the Rock-Eval® thermal analysis data of soil samples from long-term experiments where the size of the centennially stable and active SOC fractions can be estimated, and (2) apply this model on the Rock-Eval® data of unknown soils, to partition SOC into its centennially stable and active fractions. Here, we significantly extend the validity range of the machine-learning model published by Cécillon et al. [Biogeosciences, 15, 2835–2849, 2018, https://doi.org/10.5194/bg-15-2835-2018], and built upon this strategy. The second version of this statistical model, which we propose to name $PARTY_{SOC}$, uses six European long-term agricultural sites including a bare fallow treatment and one South American vegetation change ($C_4$ to $C_3$ plants) site as reference sites. The European version of the model ($PARTY_{SOC}v2.0_{EU}$) predicts the proportion of the centennially stable SOC fraction with a conservative root-mean-square error of 0.15 (relative root-mean-square error of 0.27) in a wide range of agricultural topsoils from Northwestern Europe. We plan future expansions of the $PARTY_{SOC}$ global model using additional reference soils developed under diverse pedoclimates and ecosystems, and we already recommend the application of $PARTY_{SOC}v2.0_{EU}$ in European agricultural topsoils to provide accurate information on SOC kinetic pools partitioning that may improve the simulations of simple models of SOC dynamics.





# 1 Introduction

Soil organic carbon (SOC) is identified as a key element contributing to soil functions such as primary productivity, water purification and regulation, carbon sequestration and climate regulation, habitat for biodiversity and recycling of nutrients (Keesstra et al., 2016; Koch et al., 2013; Schulte et al., 2014; Wiesmeier et al., 2019)**.** While the magnitude and the historical
dimension of the decrease in SOC at the global level are progressively being unveiled (IPBES, 2018; Sanderman et al., 2017; Stoorvogel et al., 2017), SOC stocks' preservation and even increase is a major challenge for human societies in the 21$^{st}$ century (Amundson et al., 2015). With widespread beneficial effects on soil functioning at the local level (Pellerin et al., 2019), increasing the size of the global SOC reservoir contributes directly to the Sustainable Development Goal related to life on land (https://www.globalgoals.org/15-life-on-land). It is also one of the few land management-based intervention
options that has a broad and positive impact on food security and climate change mitigation and adaptation, two other Sustainable Development Goals set by the United Nations (IPCC, 2019; Lal, 2004).

There is experimental evidence showing that in all soils, SOC is made of carbon atoms with highly contrasting residence times, ranging from hours to millennia (Balesdent et al., 1987; Trumbore et al., 1989). This continuum in SOC persistence is
often simplified by considering SOC as a mixture formed of several fractions, also called kinetic pools by modelers (Hénin and Dupuis, 1945; Jenkinson, 1990; Nikiforoff, 1936). The most drastic conceptual simplification of SOC persistence considers only two pools: (1) one made of young SOC with a short turnover rate (typically three decades on average; the active or labile SOC pool) and (2) one made of older SOC that persists much longer in the soil (more than a century; the stable, passive or persistent SOC pool). This dualistic representation of SOC persistence was considered as "*a necessary*
*simplification, but certainly not a utopian one*" four decades ago (Balesdent and Guillet, 1982) and is still considered as meaningful (*e.g.*, Lavallee et al., 2020). The active and stable soil organic matter pools contribute differently to the various soil functions (Hsieh, 1992). The active organic matter pool efficiently fuels soil biological activity (with carbon, nutrients and energy) and plant growth (with nutrients) through its rapid decay, and it sustains soil structure development (Abiven et al., 2009; Janzen, 2006). Conversely, the potential contribution of a soil to climate regulation would be most dependent on its
stable organic matter pool size (He et al., 2016; Shi et al., 2020).

A myriad of methods has been developed and tested to partition SOC into active and stable fractions, that would match kinetic pools for the assessment of SOC dynamics and related soil functions, since the second half of the 20$^{th}$ century (Balesdent, 1996; Hénin and Turc, 1949; Monnier et al., 1962; Poeplau et al., 2018). Some of these methods based on
chemical or physical (size, density or thermal) fractionation schemes can separate SOC fractions with, on average, different turnover rates (Balesdent, 1996; Plante et al., 2013; Poeplau et al., 2018; Trumbore et al., 1989). Of these methods, only a few are reasonably reproducible and easy to implement such as the ones based on rapid thermal analysis and chemical extractions (Gregorich et al., 2015; Poeplau et al., 2013, 2018; Soucémarianadin et al., 2018a). Other methods, such as size





and density SOC fractionation, need to be inferred from statistical models or infrared spectroscopy to be implemented on
large soil sample sets (Baldock et al., 2013; Cotrufo et al., 2019; Jaconi et al., 2019; Viscarra Rossel et al., 2019; Viscarra
Rossel and Hicks, 2015; Vos et al., 2018; Zimmermann et al., 2007b). However, all SOC fractionation methods fail to
achieve a proper separation of stable from active SOC, and the isolated SOC fractions are thus mixtures of centennially
stable and active SOC (Fig. 1; Balesdent, 1996; Hsieh, 1992; von Lützow et al., 2007; Sanderman and Grandy, 2020). This
limitation is common to all existing SOC fractionation methods and compromises the results of any work using them directly
to quantify soil functions specifically related to SOC fractions or to parameterize SOC partitioning in multi-compartmental
models of SOC dynamics (Luo et al., 2016). Simulations of SOC stocks changes by multi-compartmental models are very
sensitive to the initial proportion of the centennially stable SOC fraction, underlining the importance of its accurate
estimation (Clivot et al., 2019; Falloon and Smith, 2000; Jenkinson et al., 1991; Taghizadeh-Toosi et al., 2020).

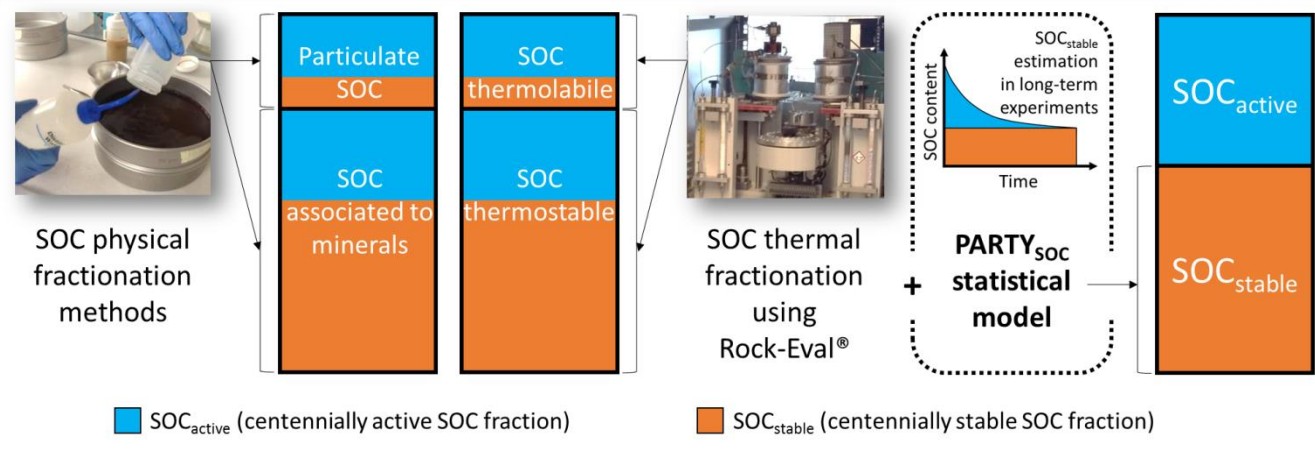


**Figure 1: Conceptual representation of soil organic carbon fractionation methods *vs*. the PARTY$_{SOC}$ approach to
quantify the size of the centennially stable and active soil organic carbon fractions.** All existing soil organic carbon
fractionation methods isolate fractions that are mixtures of centennially stable and active soil organic carbon. PARTY$_{SOC}$ is a
machine-learning model trained on the Rock-Eval® thermal analysis data of soil samples from long-term experiments where
the size of the centennially stable SOC fraction can be estimated. When applied on the Rock-Eval® data of unknown
topsoils, PARTY$_{SOC}$ partitions soil organic carbon into its active and stable fractions (*i.e.*, without isolating soil organic
carbon fractions from each other). Abbreviation: SOC, soil organic carbon. Credits for photos: SOC physical fractionation
methods, Mathilde Bryant; SOC thermal fractionation using Rock-Eval®, Lauric Cécillon.

If the stable SOC fraction cannot be isolated, it has specific chemical and thermal characteristics: stable SOC is depleted in
hydrogen and thermally stable (Barré et al., 2016; Gregorich et al., 2015). These characteristics are quickly (*ca*. 1 h per
sample) measureable using Rock-Eval® thermal analysis, and they could be of use to identify the quantitative contribution





of stable SOC to total SOC. An alternative to the elusive proper separation of stable and active SOC pools could thus be to directly predict their sizes by training a machine-learning model based on Rock-Eval® data to estimate the size of the stable

and active SOC fractions, without isolating them from each other (Fig. 1). This statistical model would need a learning set of soil samples for which SOC partitioning into its active and stable pools can be fairly estimated. Such soil samples are available in long-term (*i.e.*, at least longer than three decades) bare fallow experiments (LTBF; soils kept free of vegetation and thus with negligible SOC inputs), or long-term vegetation change ($C_3$ plants to $C_4$ plants or *vice versa*) experiments, as described by Balesdent et al. (1987, 2018), Barré et al. (2010), Cerri et al. (1985) or Rühlmann (1999). Cécillon et al. (2018)

used this strategy, developing a machine-learning random forests regression model on topsoil samples obtained from the archives of four European long-term agricultural sites including an LTBF treatment. This statistical model, which we propose to name PARTY$_{SOC}$, related thermal analysis parameters of topsoils measured with Rock-Eval® to their estimated proportion of the centennially stable SOC fraction (Fig. 1). This previous work positioned PARTY$_{SOC}$ as the first operational method quantifying the centennially stable and active SOC fractions in agricultural topsoils from Northwestern Europe.

However, the ability of this machine-learning model to fairly partition the centennially stable and the active SOC fractions of soil samples from new sites in and outside Northwestern Europe is largely unknown because its learning sample set is (1) rather limited, with a low number of reference sites and (2) based on centennially stable SOC contents that are exclusively inferred from plant-free LTBF treatments.

In this study, we aimed to improve the accuracy and the genericity of the PARTY$_{SOC}$ statistical model partitioning SOC into its centennially stable and active fractions developed by Cécillon et al. (2018). (1) We increased the range of soil types, soil texture classes, climates and types of long-term experiments, through the addition to the learning sample set of topsoils from three new reference sites (two additional European long-term agricultural sites with an LTBF treatment and one South-American long-term vegetation change site). (2) We integrated new predictor variables derived from Rock-Eval® thermal

analysis. (3) In this second version of the model, we also changed the following series of technical details. We added a new criterion based on observed SOC content to estimate of the size of the centennially stable SOC fraction at reference sites, to reduce the risk of overestimating this site-specific parameter. We calculated the proportion of the centennially stable SOC fraction differently in reference topsoil samples, using SOC content estimated by Rock-Eval® rather than by dry combustion. We changed some criteria regarding the selection of reference topsoils in the learning set of the model: we

removed samples from agronomical treatments with compost or manure amendments, and preference was given to samples with good organic carbon yield of their Rock-Eval® thermal analysis. We better balanced the contribution of each reference site to PARTY$_{SOC}$v2.0. (4) We also aimed to build a regional version of the statistical model restricted to the references sites available in Europe (named PARTY$_{SOC}$v2.0$_{EU}$). (5) Finally, we carefully evaluated the performance of the statistical models on unknown reference sites, and we further investigated the sensitivity of model performance to the reference sites included

in the learning set. For clarity, the main changes between the first version of PARTY$_{SOC}$ (Cécillon et al., 2018) and this second version of the model are summarized in supplementary Table S1.



## 2 Methods

### 2.1 Reference sites and estimation of the centennially stable SOC fraction content at each site

This second version of PARTY$_{SOC}$ uses seven long-term study sites as reference sites (*i.e.*, sites where the size of the
centennially stable SOC fraction can be estimated). The main characteristics of these seven reference sites and their
respective soil type and basic topsoil properties are presented in supplementary Table S2, and more thoroughly in the
references cited below. Six reference sites of PARTY$_{SOC}$v2.0 are long-term agricultural experiments located in Northwestern
Europe that include at least one LTBF treatment. (1) The long-term experiment on animal manure and mineral fertilizers
(B3- and B4-fields) and its adjacent LTBF experiment started in 1956 and terminated in 1985, at the Lermarken site of
Askov in Denmark (Christensen et al., 2019; Christensen and Johnston, 1997). (2) The static fertilization experiment (V120)
started in 1902 and the fallow experiment (V505a) started in 1988 at Bad Lauchstädt in Germany (Franko and Merbach,
2017; Körschens et al., 1998; Ludwig et al., 2007). (3) The "36 parcelles" experiment, started in 1959 at Grignon in France
(Cardinael et al., 2015; Houot et al., 1989). (4) The "42 parcelles" experiment, started in 1928 at Versailles in France (van
Oort et al., 2018). (5) The Highfield bare fallow experiment, started in 1959 at Rothamsted in England (Johnston et al.,
2009). (6) The Ultuna continuous soil organic matter field experiment, started in 1956 in Sweden (Kätterer et al., 2011).
These six reference sites are used in the European version of the statistical model, PARTY$_{SOC}$v2.0$_{EU}$. One additional long-
term vegetation change site completes the reference sites list of the PARTY$_{SOC}$v2.0 global statistical model. This site is a 56-
year chronosequence of oil palm plantations (with C$_3$ plants) established on former pastures (with C$_4$ plants), located in
South-America (La Cabaña in Colombia), and sampled as a space-for-time substitution (Quezada et al., 2019).


For each reference site, data on total SOC content in topsoil (0–10 cm to 0–30 cm depending on the site; supplementary
Table S2) were obtained from previously published studies (Barré et al., 2010; Cécillon et al., 2018; Franko and Merbach,
2017; Körschens et al., 1998; Quezada et al., 2019). Total SOC content was measured by dry combustion with an elemental
analyzer (SOC$_{EA}$, g C kg$^{-1}$) according to ISO 10694 (1995), after the removal of soil carbonates using an HCl treatment for
the topsoils of Grignon. For the site of La Cabaña, data on $^{13}$C content (measured using an isotope-ratio mass spectrometer
coupled to the elemental analyzer, the results being expressed in δ$^{13}$C abundance ratio (‰ relative to the international
standard)) were obtained from Quezada et al. (2019), and the relative contributions of new (C$_3$-plant derived) and old (C$_4$-
plant derived) carbon to total SOC in topsoils (0–10 cm) were calculated using the Equation 3 of the paper published by
Balesdent and Mariotti (1996), as done in Quezada et al. (2019).


Based on these published data, the content of the centennially stable SOC fraction (g C kg$^{-1}$) at each reference site was
estimated by modelling the decline of total SOC present at the onset of the experiment with time (sites with an LTBF
treatment; as SOC inputs are negligible in bare fallow systems) or by modelling the decline of C$_4$-plant derived SOC present
at the time of vegetation change with time (La Cabaña site; as SOC inputs from C$_4$ plants are negligible after pasture





conversion to oil palm plantation). For the seven reference sites, the decline in total SOC or $C_4$-plant derived SOC over time

had a similar shape, as shown in Barré et al. (2010), Cécillon et al. (2018), Franko and Merbach (2017) and Quezada et al.

(2019) and could be modelled using a first-order exponential decay with a constant term following Eq. (1):

$$\gamma(t) = ae^{-bt} + c \, , \hspace{5cm} (1)$$

where $\gamma(t)$ (g C kg$^{-1}$) is the total (sites with an LTBF treatment) or $C_4$-plant derived (La Cabaña site) SOC content at time $t$, $t$

(year) is the time under bare fallow (sites with an LTBF treatment) or since pasture conversion to oil palm plantation (La

Cabaña site), and $a$, $b$ and $c$ are fitting parameters. Parameter $a$ (g C kg$^{-1}$) corresponds to the content of the active SOC

fraction and $b$ (yr$^{-1}$) is the characteristic decay rate. The parameter $c$ (g C kg$^{-1}$) represents the content of theoretically inert

SOC. Following Barré et al. (2010), Cécillon et al. (2018) and Franko and Merbach (2017), we considered this parameter $c$

as a site-specific metric of the centennially stable SOC fraction content. As already stated in Cécillon et al. (2018), in our

view, the centennially stable SOC fraction is not biogeochemically inert; its mean age and mean residence time in soil are

both assumed to be high (centuries), though not precisely defined here. As a result, its decline with time is negligible at the

timescale of the long-term agricultural experiments or the long-term vegetation change site. We thus considered the

centennially stable SOC fraction content at each experimental site to be constant. In this study, we used the centennially

stable SOC fraction content already estimated by Franko and Merbach (2017) for the site of Bad Lauchstädt (on the LTBF

experiment started in 1988), and by Cécillon et al. (2018) for the sites of Versailles, Grignon, Rothamsted and Ultuna. We

estimated the content of the centennially stable SOC fraction for Askov and La Cabaña sites using the same Bayesian curve-

fitting method described by Cécillon et al. (2018). The Bayesian inference method was performed using Python 2.7 and the

PyMC library (Patil et al., 2010).


For the second version of PARTY$_{SOC}$, we aimed at reducing the potential bias towards an overestimation of the centennially

stable SOC fraction content at reference sites using the Eq. (1) (supplementary Table S1). This overestimation is possible at

reference sites with an LTBF treatment, as SOC inputs to bare fallow topsoils are low but not null (*e.g.*, Jenkinson and

Coleman, 1994; Petersen et al., 2005). Similarly, $C_4$-plant derived SOC inputs are possible after conversion to $C_3$ plants at

the site of La Cabaña. We thus used the lowest observed total (sites with an LTBF treatment) or $C_4$-plant derived (La Cabaña

site) topsoil SOC content value as the best estimate of the centennially stable SOC fraction content in reference sites where

this measured value was lower than the fitted value of the site-specific parameter $c$ of Eq. (1).

## 2.2 Rock-Eval® thermal analysis of topsoil samples available from reference sites

Surface soil samples (0–10 cm to 0–30 cm depending on the site; see supplementary Table S2) were obtained from the seven

reference sites described in Sect. 2.1. As described in Cécillon et al. (2018), the first version of the PARTY$_{SOC}$ statistical

model was based on a set of 118 topsoil samples corresponding to time series obtained from the soil archives of the sites of

Rothamsted (12 samples from the LTBF treatment and eight samples from the adjacent long-term grassland treatment),





Ultuna (23 samples from the LTBF treatment and 11 samples from the associated long-term cropland treatments), Grignon (12 samples from the LTBF treatment, six samples from the LTBF plus straw amendment treatment and six samples from

the LTBF plus composted straw amendment treatment) and Versailles (20 samples from the LTBF treatment and 20 samples from the LTBF plus manure amendment treatment). All 118 topsoil samples were previously analyzed using Rock-Eval® thermal analysis (Cécillon et al., 2018).

For the second version of the statistical model, 78 additional topsoil samples were provided by managers of the three new

reference sites. Thirty-five topsoil samples were obtained from the soil archives of the Askov site (19 samples corresponding to different dates of the LTBF treatment and 16 samples corresponding to different dates of the associated long-term cropland treatments). Twenty-seven topsoil samples were obtained from the soil archives of the Bad Lauchstädt site (eight samples from two dates of the mechanical LTBF treatment, eight samples from two dates of the chemical LTBF treatment and eleven samples from two dates of several long-term cropland treatments of the static fertilization experiment, eight out

of the latter coming from treatments with manure applications). Sixteen topsoil samples were obtained from the site of La Cabaña (13 samples from different $C_3$-plant oil palm fields planted at different dates and three samples from different long-term $C_4$-plant pastures).

The 78 additional topsoil samples from Askov, Bad Lauchstädt and La Cabaña were analyzed using the same Rock-Eval® 6

Turbo device (Vinci Technologies, France; see Behar et al., 2001 for a description of the apparatus) and the same setup as the one used for the sample set of the first version of the PARTY$_{SOC}$ statistical model, described by Cécillon et al. (2018). Briefly, *ca.* 60 mg of ground (< 250 µm) topsoil samples were subjected to sequential pyrolysis and oxidation phases. The Rock-Eval® pyrolysis phase was carried out in an $N_2$ atmosphere (3 min isotherm at 200 °C followed by a temperature ramp from 200 to 650 °C at a heating rate of 30 °C min$^{-1}$). The Rock-Eval® oxidation phase was carried out in laboratory air

atmosphere (1 min isotherm at 300 °C followed by a temperature ramp from 300 to 850 °C at a heating rate of 20 °C min$^{-1}$ and a final 5 min isotherm at 850 °C). Each Rock-Eval® analysis generated five thermograms corresponding to the volatile hydrocarbon effluents (HC_PYR thermogram), CO (CO_PYR thermogram) and $CO_2$ (CO2_PYR thermogram) measured at each second during the pyrolysis phase, and to the CO (CO_OX thermogram) and $CO_2$ (CO2_OX thermogram) measured at each second during the oxidation phase (Behar et al., 2001).


A series of Rock-Eval® parameters were calculated from these five thermograms. For each thermogram, five temperature parameters (all in °C) were retained: T10, T30, T50, T70 and T90, which respectively represent the temperatures corresponding to the evolution of 10, 30, 50, 70 and 90% of the total amount of evolved gas. The calculation of Rock-Eval® temperature parameters was performed using different intervals of integration depending on the thermogram. The integration

omitted the first 200 seconds of the analysis for the three thermograms of the pyrolysis phase. The integration ended at the time of analysis corresponding to the maximum oven temperatures of 650 °C (HC_PYR thermogram), 560 °C (CO_PYR





and CO2_PYR thermograms), 850 °C (CO_OX thermogram) and 611 °C (CO2_OX thermogram). These intervals of integration prevented any interference by inorganic carbon from most soil carbonates, and they ensured comparability with previous studies (Barré et al., 2016; Cécillon et al., 2018; Poeplau et al., 2019; Soucémarianadin et al., 2018b). Automatic

baseline correction (as calculated by the software of the Rock-Eval® apparatus; Vinci Technologies, France) was performed for all thermograms but the CO_PYR and the CO2_PYR thermograms. This correction can yield some negative values for the CO_PYR and CO2_PYR thermograms of soil samples with very low SOC content (data not shown). For the HC_PYR thermogram we also determined three parameters reflecting a proportion of thermally resistant or labile hydrocarbons: a parameter representing the proportion of hydrocarbons evolved between 200 and 450 °C (thermo-labile hydrocarbons,

TLHC-index, unitless; modified from Saenger et al. (2013, 2015) as described by Cécillon et al. (2018); a parameter representing the preservation of thermally labile hydrocarbons (I-index, unitless, after Sebag et al., 2016); and a parameter representing the proportion of hydrocarbons thermally stable at 400 °C (R-index, unitless, after Sebag et al., 2016). We also considered the hydrogen index (HI, mg HC $g^{-1}$ C) and oxygen index (OI$_{RE6}$, mg $O_2$ $g^{-1}$ C) that respectively describe the relative elemental hydrogen and oxygen enrichment of soil organic matter (see *e.g.*, Barré et al., 2016). These 30 Rock-

Eval® parameters are not directly related to total SOC content and were all included in the first version of the PARTY$_{SOC}$ model developed by Cécillon et al. (2018).

In this second version of PARTY$_{SOC}$, we considered ten additional Rock-Eval® parameters as possible predictors, some of these being directly linked to SOC content (supplementary Table S1). These ten parameters were calculated for all the 196

topsoil samples available from the seven reference sites. They included: the content of SOC as determined by Rock-Eval® (TOC$_{RE6}$, g C $kg^{-1}$); the content of soil inorganic carbon as determined by Rock-Eval® (MinC, g C $kg^{-1}$); the content of SOC evolved as HC, CO or $CO_2$ during the pyrolysis phase of Rock-Eval® (PC, g C $kg^{-1}$); the content of SOC evolved as HC during the temperature ramp (200–650 °C) of the pyrolysis phase of Rock-Eval® (S2, g C $kg^{-1}$); the content of SOC that evolved as HC, CO or $CO_2$ during the first 200 seconds of the pyrolysis phase (at *ca.* 200 °C) of Rock-Eval® (PseudoS1, g C

$kg^{-1}$, after Khedim et al., 2020); the ratio of PseudoS1 to PC (PseudoS1/PC, unitless); the ratio of PseudoS1 to TOC$_{RE6}$ (PseudoS1/TOC$_{RE6}$, unitless); the ratio of S2 to PC (S2/PC, unitless, after Poeplau et al., 2019); the ratio of PC to TOC$_{RE6}$ (PC/TOC$_{RE6}$, unitless); and the ratio of HI to OI$_{RE6}$ (HI/OI$_{RE6}$, mg HC $mg^{-1}$ $O_2$). TOC$_{RE6}$, MinC, PC, HI and OI$_{RE6}$ were obtained as default parameters from the software of the Rock-Eval® apparatus (Vinci Technologies, France). All other Rock-Eval® parameters were calculated from the integration of the five thermograms using R version 4.0.0 (R Core Team,

2020; RStudio Team, 2020) and functions from the R packages hyperSpec (Beleites and Sergo, 2020), pracma (Borchers, 2019) and stringr (Wickham, 2019).

### 2.3 Determination of the centennially stable SOC fraction proportion in topsoil samples from the reference sites

Following the first version of PARTY$_{SOC}$ (Cécillon et al., 2018), the proportion of the centennially stable SOC fraction in a topsoil sample of a reference site was calculated as the ratio of the site-specific centennially stable SOC fraction content (see



Sect. 2.1) to the SOC content of this particular sample. We thus assume that the centennially stable SOC fraction content in topsoils is the same in the various agronomical treatments of a reference site and that it remains constant within the time-period studied at each site.

While for the first version of PARTY$_{SOC}$, the proportion of the centennially stable SOC fraction in reference topsoils was
calculated with SOC contents determined by elemental analysis (SOC$_{EA}$), in this second version, we preferred the SOC content determined by Rock-Eval® (supplementary Table S1). The reason behind this choice was to link the Rock-Eval® parameters measured on a reference topsoil sample to a calculated proportion of the centennially stable SOC fraction that better reflected the organic carbon that actually evolved during its Rock-Eval® analysis. This choice was possible for reference topsoil samples for which Rock-Eval® analyses showed a good organic carbon yield (TOC$_{RE6}$ divided by SOC$_{EA}$,
and multiplied by 100). This is generally the case for most soils, with typical organic carbon yields of Rock-Eval® ranging from 90 to 100% of SOC$_{EA}$ (Disnar et al., 2003). For the topsoils of the sites of Grignon, Rothamsted, Ultuna and Versailles used in the first version of PARTY$_{SOC}$, the organic carbon yield of Rock-Eval® was greater than 96% (linear regression model, $R^2$ = 0.97, n = 118; Cécillon et al., 2018). Similarly, Rock-Eval® analyses of topsoil samples from the site of La Cabaña showed very good organic carbon yields (95% on average, linear regression model $R^2$ = 0.95, n = 16). For these five
reference sites (corresponding to 134 reference topsoil samples), we thus used the Rock-Eval® parameter TOC$_{RE6}$ as a measure of the SOC content of topsoil samples to calculate their respective proportion of the centennially stable SOC fraction. Conversely, Rock-Eval® analyses of topsoil samples from the sites of Askov and Bad Lauchstädt showed moderate organic carbon yields (90% on average for topsoils of Askov, with a noisy linear regression model $R^2$ = 0.68, n = 30; and 92% on average for topsoils of Bad Lauchstädt, yet with a very good linear regression model $R^2$ = 0.96, n = 11). Using the
total carbon measured by Rock-Eval® (*i.e.*, the sum of TOC$_{RE6}$ plus MinC Rock-Eval® parameters) as an estimate of the SOC content of topsoil samples for these two sites —that are not carbonated— increased the organic carbon yield of Rock-Eval® analyses (96% on average at Askov, still with a noisy linear regression model $R^2$ = 0.66, n = 30; and 101% on average at Bad Lauchstädt, with a very good linear regression model $R^2$ = 0.95, n = 11). For the two reference sites of Askov and Bad Lauchstädt (corresponding to 62 topsoil samples), we thus used the sum of Rock-Eval® parameters TOC$_{RE6}$ plus MinC as a
measure of the SOC content of topsoil samples to calculate their proportion of the centennially stable SOC fraction.

The uncertainty in the proportion of the centennially stable SOC fraction was calculated using Equation 6 of the paper published by Cécillon et al. (2018), propagating the uncertainties in SOC content data (using a standard error of 0.5 g C kg$^{-1}$, following Barré et al., 2010) and in the site-specific contents of the centennially stable SOC fraction (see above and Table 1).





**Table 1: Main statistics of soil organic carbon contents, site-specific contents of the centennially stable SOC fraction, and resulting proportions of centennially stable SOC fraction in topsoils of the seven reference sites used as the reference topsoil sample sets of the PARTY$_{SOC}$v2.0 and PARTY$_{SOC}$v2.0$_{EU}$ models.** More details on agronomical treatments and sampling year of reference topsoil samples are provided in supplementary Table S3. Abbreviations: SOC, soil organic carbon; LTBF, long-term bare fallow; min, minimum; max, maximum; sd, standard deviation.

| Reference site (Country) | Treatments (number of samples) | SOC content (g C kg$^{-1}$) mean (min, max, sd) measurement method | Centennially stable SOC fraction content (g C kg$^{-1}$) mean (sd) estimation method | Proportion of the centennially stable SOC fraction (unitless) mean (min, max, sd) |
|---|---|---|---|---|
| **Versailles** (France) | LTBF (n = 15) | 10.4 (5.6, 17.9, 3.9) TOC$_{RE6}$ | 5.50 (0.50) Lowest SOC$_{EA}$ measured on site | 0.60 (0.31, 0.98, 0.20) |
| **Rothamsted** (England) | Grassland (n = 7) LTBF (n = 8) | 28.3 (12.2, 41.5, 10.1) TOC$_{RE6}$ | 9.72 (0.50) Lowest SOC$_{EA}$ measured on site | 0.40 (0.23, 0.80, 0.18) |
| **Ultuna** (Sweden) | Cropland (n = 3; +straw n = 8) LTBF (n = 4) | 15.2 (10.0, 20.3, 2.8) TOC$_{RE6}$ | 6.95 (0.88) Bayesian curve-fitting | 0.47 (0.34, 0.70, 0.09) |
| **Grignon** (France) | LTBF (n = 12, +straw n = 3) | 11.5 (8, 14.3, 1.7) TOC$_{RE6}$ | 7.12 (1.00) Bayesian curve-fitting | 0.63 (0.50, 0.89, 0.10) |
| **Askov** (Denmark) | Cropland (n = 7) LTBF (n = 8) | 13.8 (11.1, 16.8, 1.9) TOC$_{RE6}$+MinC | 5.10 (0.88) Bayesian curve-fitting | 0.38 (0.30, 0.46, 0.05) |
| **Bad Lauchstädt** (Germany) | Cropland (n = 1) LTBF (n = 14) | 18.0 (16.8, 19.4, 0.6) TOC$_{RE6}$+MinC | 15.00 (0.50) Lowest SOC$_{EA}$ measured on site | 0.84 (0.77, 0.89, 0.03) |
| **La Cabaña** (Colombia) | Pasture (n = 3) Oil-palm plantation (n = 12) | 17.8 (10.2, 31.8, 5.7) TOC$_{RE6}$ | 4.75 (0.50) Lowest SOC$_{EA}$ measured on site | 0.29 (0.15, 0.47, 0.10) |
| **Reference soil sample set of PARTY$_{SOC}$v2.0** (n = 105) | | 16.4 (5.6, 41.5, 7.3) | | 0.52 (0.15, 0.98, 0.21) |
| **Reference soil sample set of PARTY$_{SOC}$v2.0$_{EU}$** (n = 90) | | 16.2 (5.6, 41.5, 7.5) | | 0.55 (0.23, 0.98, 0.20) |



**2.4 Selection of the learning set and of meaningful Rock-Eval® predictors variables for the PARTY$_{SOC}$v2.0 model**

In machine-learning, the selection of the learning set (here, the training and test sets of reference topsoil samples) of the
model influences the performances of the model, just like the selection of the predictor variables (here, the Rock-Eval®
parameters) (*e.g.*, Cécillon et al., 2008; Wehrens, 2020).

For this second version of PARTY$_{SOC}$, we changed some criteria regarding the inclusion of the available reference topsoil
samples in the learning set of the model (supplementary Table S1). We excluded from the learning set all the topsoil samples
experiencing agronomical treatments that may have changed the site-specific content of the centennially stable SOC fraction.
These agronomical treatments concern the repeated application of some types of exogenous organic matter such as compost
or manure, for which we suspect that they may increase the content of the centennially stable SOC fraction after several
decades. Therefore, to increase the likelihood of verifying our hypothesis of a constant content of the centennially stable
SOC fraction at each reference site in time and space (see Sect. 2.3), we excluded all reference topsoil samples experiencing
repeated applications of composted straw (six samples from Grignon), or manure (20 samples from Versailles and eight
samples from Bad Lauchstädt) from the learning set of the statistical model. Yet, we kept some reference topsoil samples
from Grignon and Ultuna experiencing repeated applications of straw.

We also excluded from the learning set of the model the reference topsoil samples for which the organic carbon yield of
Rock-Eval® is below 86% or above 116%. For the site of Askov with a noisy relationship between SOC$_{EA}$ and the sum
TOC$_{RE6}$ plus MinC (see Sect. 2.3), we excluded the five samples without a SOC$_{EA}$ measurement preventing the calculation
of the organic carbon yield of their Rock-Eval® analysis. Conversely, for the site of Bad Lauchstädt we kept topsoil samples
without available SOC$_{EA}$ measurements, as the linear relationship between SOC$_{EA}$ and the sum TOC$_{RE6}$ plus MinC was very
good for this site (see Sect. 2.3). These criteria regarding the organic carbon yield of Rock-Eval® lead to the exclusion of
nine samples from the site of Askov, four additional samples from the site of Versailles and two from the site of Ultuna.

Contrary to the first version of PARTY$_{SOC}$ (Cécillon et al., 2018), this second version is based on a balanced contribution of
each reference site to the statistical model (supplementary Table S1). Each reference site contributes to the model with 15
samples, so that the reference sample set of the PARTY$_{SOC}$v2.0 statistical model is composed of 105 topsoil samples (90 for
the European version of the model PARTY$_{SOC}$v2.0$_{EU}$). Besides the above-mentioned exclusion criteria (that excluded 49 out
of the 196 topsoil samples available from the seven reference sites), the 15 topsoil samples retained for each reference site
were selected: (1) to have a range of proportion of centennially stable SOC fraction as wide as possible; (2) to have the best
organic carbon yield of Rock-Eval® analysis. On average, the organic carbon yield of the Rock-Eval® analyses for the
retained learning set of reference topsoil samples (calculated as described above) was greater than 98% of SOC$_{EA}$
(SOC$_{DETERMINED\_BY\_ROCK\text{-}EVAL®}$ = 0.9924 SOC$_{EA}$ - 0.1051, R² = 0.99, n = 91 topsoil samples with available SOC$_{EA}$





measurements). The list of the 105 reference topsoil samples retained as the learning set of PARTY$_{SOC}$v2.0 is provided in supplementary Table S3. This list includes, for each reference topsoil sample, information on its reference site, land cover, agronomical treatment, sampling year and its values for the 40 Rock-Eval® parameters.

The 40 Rock-Eval® parameters calculated (see Sect. 2.2) captured most of the information related to SOC thermal stability, elemental stoichiometry and content that is contained in the five Rock-Eval® thermograms. However, not all Rock-Eval® parameters do necessarily carry meaningful information for partitioning SOC into its centennially stable and active fractions (Cécillon et al., 2018). The PARTY$_{SOC}$v2.0 statistical model and its European version PARTY$_{SOC}$v2.0$_{EU}$ incorporate as predictor variables only the Rock-Eval® parameters showing a strong relationship with the proportion of the centennially

stable SOC fraction (supplementary Table S1). The absolute value of 0.50 for the Spearman's rho (nonparametric and nonlinear correlation test) was used as a threshold to select meaningful Rock-Eval® predictor variables (calculated on the reference topsoil sample set of the PARTY$_{SOC}$v2.0 model, n = 105). Basic statistics of all Rock-Eval® parameters (learning set of PARTY$_{SOC}$v2.0) are reported in supplementary Table S4.

## 2.5 Random forests regression models to predict the proportion of the centennially stable SOC fraction from Rock-
Eval® parameters, performance assessment and error propagation in the statistical models

The PARTY$_{SOC}$v2.0 statistical model consists of a nonparametric and nonlinear multivariate regression model relating the proportion of the centennially stable SOC fraction (response vector or dependent variable **y**) of the reference soil sample set (n = 105 topsoil samples from the seven reference sites, see Sect. 2.4) to their Rock-Eval® parameters summarized by a matrix of predictor variables (**X**) made up of the selected centered and scaled Rock-Eval® parameters. As stated above, we

also built a regional (European) version of the statistical model based on the six European reference sites only (PARTY$_{SOC}$v2.0$_{EU}$, using the 90 reference topsoil samples from Askov, Bad Lauchstädt, Grignon, Rothamsted, Ultuna and Versailles).

Like the first version of the PARTY$_{SOC}$ statistical model, this second version uses the machine-learning algorithm of random
forests-random inputs (hereafter termed random forests) proposed by Breiman (2001). This algorithm aggregates a collection of random regression trees (Breiman, 2001; Genuer and Poggi, 2020). The PARTY$_{SOC}$v2.0 and its European version PARTY$_{SOC}$v2.0$_{EU}$ are based on a forest of 1000 different regression trees made of splits and nodes. The learning algorithm of random forests combines bootstrap resampling and random variable selection. Each of the 1000 regression trees was grown on a bootstrapped subset of the reference topsoil sample set (*i.e.*, containing *ca.* two-thirds of "in-bag" samples). The

algorithm randomly sampled one-third out of the selected Rock-Eval® parameters (see Sect. 2.4) as candidates at each split of the regression tree, and it used a minimum size of terminal tree nodes of five topsoil samples. The relative importance (*i.e.*, ranking) of each selected Rock-Eval® parameters in the regression models was computed as the unscaled permutation accuracy (Strobl et al., 2009).





The performance of the $PARTY_{SOC}v2.0$ and the $PARTY_{SOC}v2.0_{EU}$ random forests regression models was assessed by statistical metrics comparing the predicted *vs.* the estimated values of their reference topsoil sample set using three different strategies. First, the predictive ability of both models was assessed by an "internal" procedure that used their respective whole reference topsoil sample sets (n = 105 samples for $PARTY_{SOC}v2.0$, n = 90 samples for $PARTY_{SOC}v2.0_{EU}$). For this procedure, performance statistics were calculated only on the "out-of-bag" topsoil samples of the whole reference sets, using

a random seed of 1 to initialize the pseudorandom number generator of the R software. Out-of-bag samples are observations from the training sets not included in the learning topsoil sample set for a specific regression tree that can be used as a "built-in" test set for calculating its prediction accuracy (Strobl et al., 2009). Second, the predictive ability of the models was assessed by a "random splitting" procedure that split randomly their respective reference topsoil sample sets into a test set (made of n = 30 samples), and a training set (n = 75 samples for $PARTY_{SOC}v2.0$, n = 60 samples for $PARTY_{SOC}v2.0_{EU}$). This

procedure was repeated 15 times using random seeds from 1 to 15 in the R software. Third, a fully independent "leave-one-site-out" procedure was used to assess the predictive ability of the models. This procedure successively excluded topsoil samples of one reference site from the training set and uses them as a test set (n = 15) for the models. It used the random seed of 1 in the R software. For the second and third procedures, performance statistics were calculated (1) on the "out-of-bag" topsoil samples of the training sets and (2) on the topsoil samples of the test sets.


Finally, the sensitivity of model performance to the reference sites included in the learning set of the random forests regression model was assessed on independent soils from two reference sites, used as examples. For this sensitivity analysis, topsoil samples from Grignon and Versailles (n = 15 samples) were successively used as fully independent test sets for several random forests regression models. Combinations of topsoil samples from a decreasing number of the remaining

reference sites were selected as training sets for the models, on the basis of their potential proximity to the topsoil samples of the test sets, regarding their pedological or climatic conditions. The size of the various training sets composed for the sensitivity analysis ranged from n = 90 samples (six training reference sites) to n = 30 samples (only two training reference sites).

Several statistics were used to assess the predictive ability of the regression models. The coefficient of determination: $R^2_{OOB}$, calculated on the "out-of-bag" samples of the training sets; and $R^2$, calculated on the samples of the test sets. The root-mean-square error of prediction: $RMSEP_{OOB}$, calculated on the "out-of-bag" samples of the training sets; and RMSEP, calculated on the samples of the test sets. The relative RMSEP: $_RRMSEP$, calculated as the ratio of the RMSEP to the mean value of the test sets. The ratio of performance to interquartile range (RPIQ) was calculated as the ratio of the interquartile range of the

test sets (Q3 - Q1; which gives the range accounting for 50% of the test sets around its median value) to the RMSEP (Bellon-Maurel et al., 2010). The bias of the random forests regression models was calculated as the mean of the model predictions on the test sets minus the actual mean of the test sets. Additionally, site-specific RMSEP and $_RRMSEP$ were calculated for





the "leave-one-site-out" procedure (on the 15 independent test topsoil samples from each site). The uncertainty on the model predictions for new topsoils was determined using a methodology that was fully described by Cécillon et al. (2018). This

methodology was adapted after the work of Coulston et al. (2016), to explicitly take into account the uncertainty in the reference values of the proportion of the centennially stable SOC fraction (see Sect. 2.3) that were used to build the models (Cécillon et al., 2018).

PARTY$_{SOC}$v2.0 and PARTY$_{SOC}$v2.0$_{EU}$ were programmed as R scripts in the RStudio environment software (RStudio Team,

2020), and were run using the R version 4.0.0 (R Core Team, 2020). The R scripts use the random forests algorithm of the randomForest R package (Liaw and Wiener, 2002) and the boot R package for bootstrapping (Canty and Ripley, 2020; Davison and Hinkley, 1997).

## 3 Results

### 3.1 Content of the centennially stable SOC fraction at the reference sites

The two newly fitted values of the centennially stable SOC fraction content (*i.e.*, parameter $c$ in Eq. (1), see Sect. 2.1) were 5.10 g C kg$^{-1}$ at the site of Askov (standard deviation = 0.88 g C kg$^{-1}$) and 5.12 g C kg$^{-1}$ at the site of La Cabaña (standard deviation = 0.35 g C kg$^{-1}$). The fitted values of parameter $c$ in Eq. (1) for all reference sites and their standard errors are provided in supplementary Table S2. A total (reference sites with an LTBF treatment) or a C$_4$-plant derived (La Cabaña site) SOC content value lower than the fitted value of the site-specific parameter $c$ in Eq. (1) was measured in four out the seven

reference sites of the PARTY$_{SOC}$v2.0 model. At Bad Lauchstädt, a SOC$_{EA}$ value of 15.0 g C kg$^{-1}$ was reported by Körschens et al. (1998) for topsoils of the well ring experiment (Ansorge, 1966). At Rothamsted, a SOC$_{EA}$ measurement of 9.72 g C kg$^{-1}$ was reported for topsoils of the Highfield LTBF experiment by Cécillon et al. (2018). At Versailles a SOC$_{EA}$ measurement of 5.50 g C kg$^{-1}$ was reported after 80 years of bare fallow by Barré et al. (2010). At La Cabaña, a C$_4$-plant derived SOC content of 4.75 g C kg$^{-1}$ was calculated using data from Quezada et al. (2019). These values were thus retained as the best

estimates of the site-specific content of the centennially stable SOC fraction in topsoils of the four sites (Table 1). As these site-specific values of the centennially stable SOC fraction content were derived from SOC$_{EA}$ measurements, we attributed a standard deviation of 0.50 g C kg$^{-1}$ to each of them, following Barré et al. (2010). The final estimates of the content of the centennially stable SOC fraction at the seven reference sites that were used in the PARTY$_{SOC}$v2.0 statistical model are provided in Table 1. They varied by a factor of three across the reference sites, ranging from 4.75 g C kg$^{-1}$ at La Cabaña to

15.00 g C kg$^{-1}$ at Bad Lauchstädt. The lowest value of the topsoil content of the centennially stable SOC fraction used in the European version PARTY$_{SOC}$v2.0$_{EU}$ of the statistical model differed only slightly from the one of the PARTY$_{SOC}$v2.0 model (5.10 g C kg$^{-1}$ at the site of Askov).





**3.2 Content and biogeochemical stability of SOC in the learning sets, and selection of meaningful Rock-Eval®
parameters as predictor variables for the PARTY$_{SOC}$v2.0 and PARTY$_{SOC}$v2.0$_{EU}$ models**

The SOC content in the topsoil samples of the seven reference sites ranged from 5.6 to 41.5 g C kg$^{-1}$ in the learning sets of
the PARTY$_{SOC}$v2.0 (n = 105) and PARTY$_{SOC}$v2.0$_{EU}$ (n = 90) models (Table 1). As showed in Table 1, this resulted in
proportions of the centennially stable SOC fraction ranging from 0.15 to 0.98 (PARTY$_{SOC}$v2.0 learning set), and from 0.23
to 0.98 (PARTY$_{SOC}$v2.0$_{EU}$ learning set). All the 25 calculated Rock-Eval® temperature parameters showed positive values of
Spearman's rho coefficient with the response variable of the PARTY$_{SOC}$v2.0 model (n = 105; with Spearman's rho values up

to 0.81 for T90$_{HC\_PYR}$; Table 2). While the inorganic carbon content was not correlated to the proportion of the centennially
stable SOC fraction, TOC$_{RE6}$ was significantly and negatively correlated to the response variable of the PARTY$_{SOC}$v2.0
model (Spearman's rho = -0.55; Table 2). Other Rock-Eval® parameters linked to soil carbon content showed a stronger
relationship than TOC$_{RE6}$ with the proportion of the centennially stable SOC fraction. This was the case for S2 and PC that
showed the highest absolute Spearman's rho coefficients, with a highly significant negative relationship (Spearman's rho = -

0.85; Table 2). Eighteen out of the 40 calculated Rock-Eval® parameters showed an absolute value of Spearman's rho above
0.5 with the proportion of the centennially stable SOC fraction in the learning set of the PARTY$_{SOC}$v2.0 model (n = 105;
Table 2), and were thus retained as predictor variables for the models. The 18 Rock-Eval® parameters retained were: the
Rock-Eval® temperature parameters T70$_{HC\_PYR}$, T90$_{HC\_PYR}$, T30$_{CO2\_PYR}$, T50$_{CO2\_PYR}$, T70$_{CO2\_PYR}$, T90$_{CO2\_PYR}$, T70$_{CO\_OX}$,
T50$_{CO2\_OX}$, T70$_{CO2\_OX}$, T90$_{CO2\_OX}$, and the Rock-Eval® parameters PseudoS1, S2, S2/PC, HI, HI/OI$_{RE6}$, PC, PC/TOC$_{RE6}$, and

TOC$_{RE6}$.

**Table 2: Spearman's rank correlation coefficient test between the 40 calculated Rock-Eval® parameters and the
proportion of the centennially stable organic carbon fraction in the reference topsoil sample set of the PARTY$_{SOC}$v2.0
model (n = 105), and variable importance (ranking) of the 18 selected Rock-Eval® parameters for predicting the**

**proportion of the centennially stable SOC fraction in the PARTY$_{SOC}$v2.0 and PARTY$_{SOC}$v2.0$_{EU}$ random forests
regression models.** Symbols for *p*-values: *** $p < 0.001$; ** $p < 0.01$; * $p < 0.05$; NS $p > 0.05$ = not significant. See Section
2.2 for a description of the units of the 40 Rock-Eval® parameters. The 18 Rock-Eval® parameters retained as predictor
variables for the second version of PARTY$_{SOC}$ are shown in bold. Abbreviation: SOC, soil organic carbon.

| Rock-Eval® parameter | Spearman's ρ with the proportion of the centennially stable SOC fraction | *p*-value | Variable importance to predict the proportion of the centennially stable SOC fraction in the PARTY$_{SOC}$v2.0 regression model (rank) | Variable importance to predict the proportion of the centennially stable SOC fraction in the PARTY$_{SOC}$v2.0$_{EU}$ regression model (rank) |
|---|---|---|---|---|
| T10$_{HC\_PYR}$ | 0.38 | 0.0001 | NA | NA |





| | | | | |
|---|---|---|---|---|
| $T30_{HC\_PYR}$ | 0.47 | 0.0000 | NA | NA |
| $T50_{HC\_PYR}$ | 0.46 | 0.0000 | NA | NA |
| **$T70_{HC\_PYR}$** | **0.54** | **0.0000** | **17** | **15** |
| **$T90_{HC\_PYR}$** | **0.81** | **0.0000** | **5** | **13** |
| $T10_{CO\_PYR}$ | 0.40 | 0.0000 | NA | NA |
| $T30_{CO\_PYR}$ | 0.36 | 0.0001 | NA | NA |
| $T50_{CO\_PYR}$ | 0.33 | 0.0005 | NA | NA |
| $T70_{CO\_PYR}$ | 0.31 | 0.0014 | NA | NA |
| $T90_{CO\_PYR}$ | 0.31 | 0.0013 | NA | NA |
| $T10_{CO2\_PYR}$ | 0.35 | 0.0003 | NA | NA |
| **$T30_{CO2\_PYR}$** | **0.56** | **0.0000** | **12** | **10** |
| **$T50_{CO2\_PYR}$** | **0.55** | **0.0000** | **8** | **2** |
| **$T70_{CO2\_PYR}$** | **0.55** | **0.0000** | **10** | **7** |
| **$T90_{CO2\_PYR}$** | **0.58** | **0.0000** | **11** | **11** |
| $T10_{CO\_OX}$ | 0.31 | 0.0013 | NA | NA |
| $T30_{CO\_OX}$ | 0.41 | 0.0000 | NA | NA |
| $T50_{CO\_OX}$ | 0.49 | 0.0000 | NA | NA |
| **$T70_{CO\_OX}$** | **0.58** | **0.0000** | **9** | **16** |
| $T90_{CO\_OX}$ | 0.33 | 0.0007 | NA | NA |
| $T10_{CO2\_OX}$ | 0.10 | 0.3349 | NA | NA |
| $T30_{CO2\_OX}$ | 0.39 | 0.0000 | NA | NA |
| **$T50_{CO2\_OX}$** | **0.63** | **0.0000** | **13** | **14** |
| **$T70_{CO2\_OX}$** | **0.70** | **0.0000** | **4** | **12** |
| **$T90_{CO2\_OX}$** | **0.60** | **0.0000** | **14** | **17** |
| I-index | -0.40 | 0.0000 | NA | NA |
| R-index | 0.47 | 0.0000 | NA | NA |
| TLHC-index | -0.49 | 0.0000 | NA | NA |
| **HI** | **-0.72** | **0.0000** | **7** | **6** |
| $OI_{RE6}$ | -0.09 | 0.3504 | NA | NA |
| **$TOC_{RE6}$** | **-0.55** | **0.0000** | **6** | **9** |
| MinC | 0.03 | 0.7430 | NA | NA |
| **PC** | **-0.85** | **0.0000** | **2** | **3** |
| **S2** | **-0.85** | **0.0000** | **1** | **1** |





| | | | | |
|---|---|---|---|---|
| **PseudoS1** | **-0.50** | **0.0000** | **18** | **18** |
| PseudoS1/PC | 0.28 | 0.0033 | NA | NA |
| PseudoS1/TOC$_{RE6}$ | -0.06 | 0.5702 | NA | NA |
| **S2/PC** | **-0.70** | **0.0000** | **16** | **4** |
| **PC/TOC$_{RE6}$** | **-0.71** | **0.0000** | **3** | **8** |
| **HI/OI$_{RE6}$** | **-0.68** | **0.0000** | **15** | **5** |

### 3.3 Performance assessment of the PARTY$_{SOC}$v2.0 and PARTY$_{SOC}$v2.0$_{EU}$ statistical models

Using both the "internal" and the "random splitting" performance assessment procedures (see Sect. 2.5), the PARTY$_{SOC}$v2.0 and PARTY$_{SOC}$v2.0$_{EU}$ models showed good to very good predictive ability of the proportion of the centennially stable SOC fraction (Fig. 2a; Table 3a). For most of the calculated statistics, the European version of the model PARTY$_{SOC}$v2.0$_{EU}$ showed better performances than the PARTY$_{SOC}$v2.0 model (Table 3). Using the "random splitting" procedure, the mean $R^2$
of PARTY$_{SOC}$v2.0$_{EU}$ was 0.87 (0.81 for PARTY$_{SOC}$v2.0), its RMSEP and $_R$RMSEP were respectively 0.07 and 0.13 (0.09 and 0.17 for PARTY$_{SOC}$v2.0), and its mean RPIQ was 4.6 (3.6 for PARTY$_{SOC}$v2.0). The bias was low for both models (Table 3a).

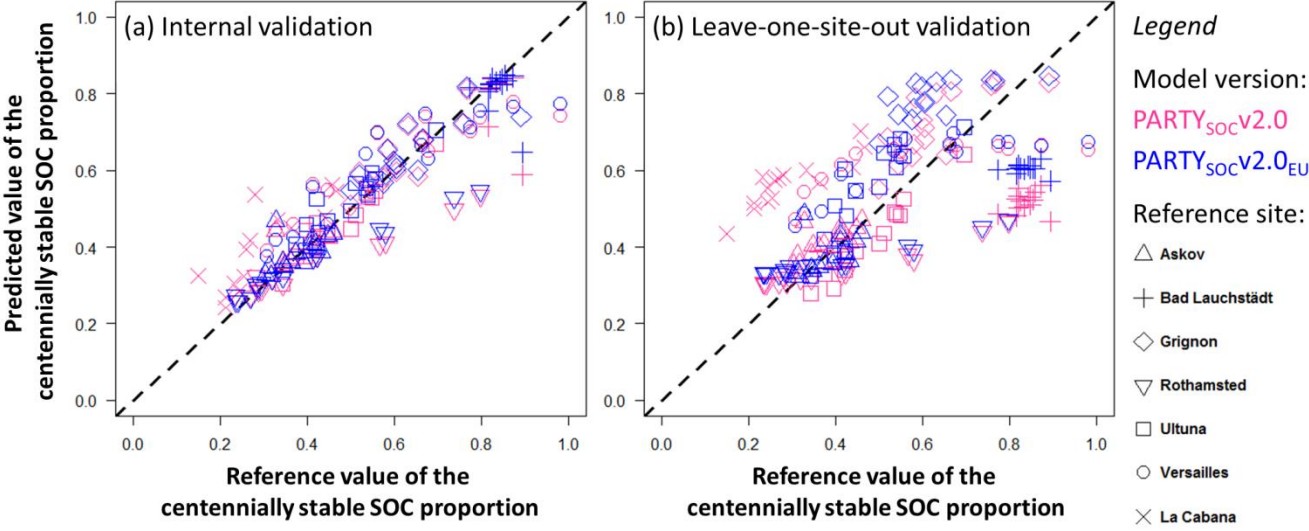

**Figure 2: Performance of the PARTY$_{SOC}$v2.0 and the PARTY$_{SOC}$v2.0$_{EU}$ statistical models based on Rock-Eval® thermal analysis for predicting the centennially stable organic carbon proportion in topsoils.** (a) Results of the internal validation procedure; (b) Results of the leave-one-site-out validation procedure (see Section 2.5 for more details on model performance assessment). Abbreviation: SOC, soil organic carbon.





**Table 3: Performance of the PARTY$_{SOC}$v2.0 and the PARTY$_{SOC}$v2.0$_{EU}$ random forests regression models based on** **Rock-Eval® thermal analysis for predicting the proportion of the centennially stable organic carbon fraction in** **topsoils.** (a) Performance statistics calculated for the internal, random splitting (mean statistics of 15 different models) and leave-one-site-out validation procedures; (b) Site-specific performance statistics calculated for the leave-one-site-out validation procedure. The performance statistics and their abbreviations are defined at Section 2.5.

| (a) | Internal procedure | | Random splitting procedure | | Leave-one-site-out procedure | |
|---|---|---|---|---|---|---|
| | PARTY$_{SOC}$v2.0 | PARTY$_{SOC}$v2.0$_{EU}$ | PARTY$_{SOC}$v2.0 | PARTY$_{SOC}$v2.0$_{EU}$ | PARTY$_{SOC}$v2.0 | PARTY$_{SOC}$v2.0$_{EU}$ |
| $R^2_{OOB}$ | 0.83 | 0.87 | 0.80 | 0.84 | - | - |
| RMSEP$_{OOB}$ | 0.08 | 0.07 | 0.09 | 0.08 | - | - |
| $R^2$ | - | - | 0.81 | 0.87 | 0.23 | 0.45 |
| RMSEP | - | - | 0.09 | 0.07 | 0.18 | 0.15 |
| $_R$RMSEP | - | - | 0.17 | 0.13 | 0.36 | 0.27 |
| RPIQ | - | - | 3.59 | 4.60 | 1.75 | 2.39 |
| Bias | - | - | 0.005 | 0.006 | < 0.001 | -0.003 |

| (b) | | Leave-one-site-out procedure | | | | | | |
|---|---|---|---|---|---|---|---|---|
| Test set | | Askov | Bad Lauchstädt | Grignon | Versailles | Rothamsted | Ultuna | La Cabaña |
| PARTY$_{SOC}$v2.0 | Site-specific RMSEP | 0.05 | 0.32 | 0.11 | 0.17 | 0.14 | 0.06 | 0.28 |
| | Site-specific $_R$RMSEP | 0.13 | 0.38 | 0.18 | 0.28 | 0.36 | 0.13 | 0.94 |
| PARTY$_{SOC}$v2.0$_{EU}$ | Site-specific RMSEP | 0.05 | 0.23 | 0.18 | 0.14 | 0.14 | 0.09 | - |
| | Site-specific $_R$RMSEP | 0.13 | 0.28 | 0.28 | 0.24 | 0.35 | 0.20 | - |



The predictive ability of both models decreased when assessed using the "leave-one-site-out" procedure (see Sect. 2.5; Fig. 2b). Again, PARTY$_{SOC}$v2.0$_{EU}$ showed better performance statistics than the PARTY$_{SOC}$v2.0 model (Table 3; Fig. 2b), with an R² of 0.45, an RMSEP of 0.15, an $_R$RMSEP of 0.27 and an RPIQ of 2.4. The PARTY$_{SOC}$v2.0 model poorly predicted the
proportion of the centennially stable SOC fraction in topsoil samples of two sites (Table 3b; Fig. 2b): La Cabaña (overestimation; with a site-specific RMSEP of 0.28) and Bad Lauchstädt (underestimation; with a site-specific RMSEP of 0.32). The proportion of the centennially stable SOC fraction in topsoil samples of Bad Lauchstädt remained underestimated by the PARTY$_{SOC}$v2.0$_{EU}$ model, though with a reduced site-specific RMSEP (0.23; Table 3b; Fig. 2b). All other site-specific RMSEPs were below 0.18 (0.17 at Versailles for PARTY$_{SOC}$v2.0, 0.18 at Grignon for PARTY$_{SOC}$v2.0$_{EU}$; Table 3b), with
remarkably low site-specific RMSEPs for the sites of Askov (below 0.05 for both models) and Ultuna (0.06 for PARTY$_{SOC}$v2.0; 0.09 for PARTY$_{SOC}$v2.0$_{EU}$).

The most important Rock-Eval® parameter for predicting the proportion of the centennially stable SOC fraction is S2 for both PARTY$_{SOC}$v2.0 and PARTY$_{SOC}$v2.0$_{EU}$ statistical models (Table 2). Conversely, the two models show only two Rock-
Eval® parameters in common out of their five most important ones that are S2, PC, PC/TOC$_{RE6}$, T70$_{CO2\_OX}$, T90$_{HC\_PYR}$ for PARTY$_{SOC}$v2.0 and S2, T50$_{CO2\_PYR}$, PC, S2/PC, HI/OI$_{RE6}$ for PARTY$_{SOC}$v2.0$_{EU}$ (Table 2).

### 3.4 Sensitivity of model performance to the reference sites included in the learning set

Restricting the learning set of the machine-learning model to topsoil samples from fewer reference sites with pedoclimatic conditions closer to the ones of a fully independent test site changed its performances (Fig. 3). Removing the reference sites
with a climate (*i.e.*, La Cabaña) or a soil type (*i.e.*, Bad Lauchstädt) differing strongly from the independent test site (here, Grignon or Versailles used as examples) reduced the site-specific RMSEP and $_R$RMSEP of the model (supplementary Table S5). When Grignon or Versailles were used as independent test sites, the statistical model with the best predictive ability (*i.e.*, the lowest site-specific RMSEP and $_R$RMSEP) used a learning set composed of 45 topsoil samples from three European reference sites (including the French site with the closest climate, despite its different soil type; supplementary Table S2 and
S5; Fig. 3).



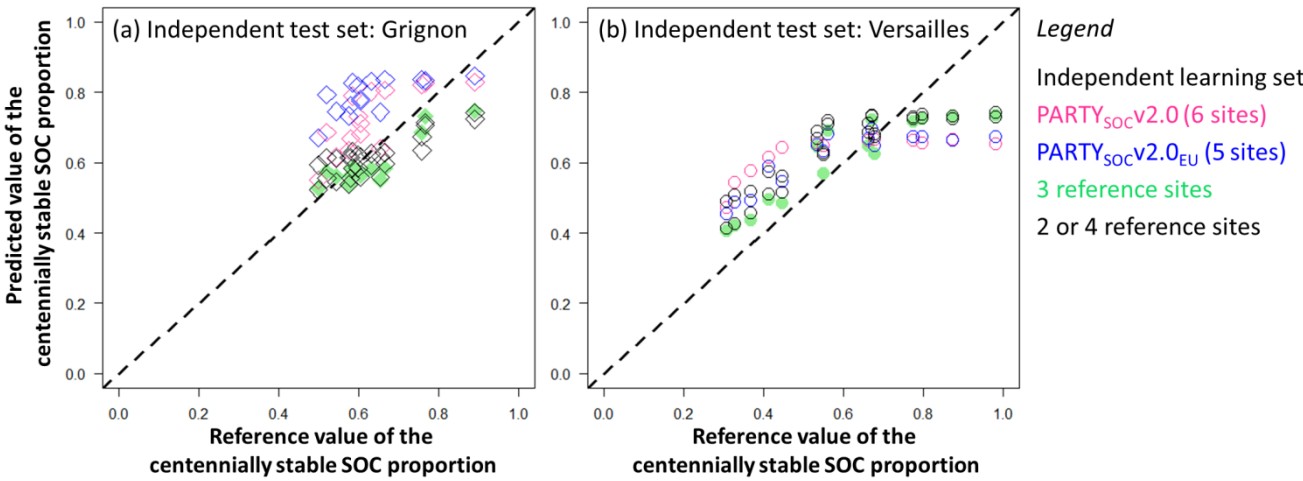

**Figure 3: Sensitivity of model performance to the reference sites included in the learning set, using 15 topsoil samples from the sites of (a) Grignon or (b) Versailles as independent test sets.** Predictions by statistical models showing the lowest RMSEP and $_R$RMSEP are plotted in green (using a learning set composed of three independent reference sites to predict Grignon or Versailles as test set). See supplementary Table S5 for more details on the learning sets of the different statistical models and their site-specific performance statistics. Abbreviation: SOC, soil organic carbon.




## 4 Discussion

The second version of the PARTY$_{SOC}$ model incorporates a large number of modifications and improvements (supplementary Table S1), and its predictive ability was more thoroughly assessed compared to the first version of the statistical model (Cécillon et al., 2018). The critical examination of the performance of PARTY$_{SOC}$v2.0 and
PARTY$_{SOC}$v2.0$_{EU}$ provides new insights: (1) on the relationships between Rock-Eval® parameters and the century-scale persistence of SOC; (2) on both current and potential capabilities of the model to partition the centennially stable and active organic carbon fraction in topsoils. Based on those insights, (3) we plan future expansions of the PARTY$_{SOC}$ global model, and we recommend the application of PARTY$_{SOC}$v2.0$_{EU}$ in European agricultural topsoils to provide accurate information on SOC kinetic pools partitioning that may improve the simulations of simple models of SOC dynamics.

**4.1 Rock-Eval® chemical and thermal information are related to the century-scale persistence of SOC**

The methodology used to estimate the centennially stable SOC proportion in reference topsoils has been revised for the second version of the PARTY$_{SOC}$ model (see Sect. 2.1 and 2.3 and supplementary Table S1), and the learning set now integrates a wider range of centennially stable SOC contents [4.75–15.00 g C kg$^{-1}$] with a median value of 6.95 g C kg$^{-1}$ (n = 7; Table 1). This range covers most of the published size estimates of this fraction in topsoils, estimated using different
methods (Balesdent et al., 1988; Barré et al., 2010; Buyanovsky and Wagner, 1998b; Cécillon et al., 2018; Franko and Merbach, 2017; Hsieh, 1992; Huggins et al., 1998; Jenkinson and Coleman, 1994; Körschens et al., 1998; Rühlmann, 1999). The contribution of each reference site to the learning set and the inclusion criteria for topsoil samples were also modified, and ten Rock-Eval® parameters not considered in the first version of the model were proposed as potential predictor variables for this second version of the statistical model (see Sect. 2.2 and 2.4 and supplementary Table S1).

Using this improved design, all Rock-Eval® temperature parameters showed positive values of Spearman's rho coefficient with the proportion of the centennially stable SOC fraction in topsoils (Table 2), when a few of them showed counterintuitive significant negative correlations using the learning set of the first version of PARTY$_{SOC}$ (Cécillon et al., 2018). This confirms the generic link between SOC thermal stability and its *in situ* biogeochemical stability: centennially
stable SOC is thermally stable, even though thermostable SOC fractions are a mixture of centennially stable and active SOC (Fig. 1; Barré et al., 2016; Gregorich et al., 2015; Plante et al., 2013; Sanderman and Grandy, 2020; Schiedung et al., 2017). Some Rock-Eval® temperature parameters were within the five most important predictor variables for both PARTY$_{SOC}$v2.0 (T70$_{CO2\_OX}$, T90$_{HC\_PYR}$) and PARTY$_{SOC}$v2.0$_{EU}$ (T50$_{CO2\_PYR}$) statistical models (Table 2).

Contrary to the first version of the PARTY$_{SOC}$ statistical model, the second version tested several Rock-Eval® parameters directly linked to soil carbon content as potential predictor variables. TOC$_{RE6}$ was selected as a meaningful predictor variable for PARTY$_{SOC}$v2.0 and PARTY$_{SOC}$v2.0$_{EU}$. Its negative correlation with the centennially stable SOC proportion (Table 2) was





expected, according to the calculation of the latter (see Sect. 2.3). This is in line with results from SOC-dating techniques and with most multi-compartmental models of SOC dynamics suggesting that the proportion of the most persistent SOC

fraction is a decreasing function of total SOC (Huggins et al., 1998; Rühlmann, 1999). Indeed, the *ex-post* optimized initial value of the proportion of the inert SOC fraction for the simple AMG model of SOC dynamics is higher (0.60 on average) for SOC-poor temperate topsoils with a long-term arable history than for SOC-rich temperate topsoils with a long-term grassland history (0.47 on average; Clivot et al., 2019). Contrarily, the empirical function commonly used to initialize the size of the inert SOC fraction of the multi-compartmental RothC model predicts an increased proportion of inert SOC with

increased total SOC (Falloon et al., 1998). This empirical function needs to be examined upon these results.

Interestingly, S2 (pyrolysable volatile hydrocarbon effluents) and PC (total pyrolysable organic carbon), two other Rock-Eval® parameters linked to SOC content showed a stronger negative relationship than $TOC_{RE6}$ with the proportion of the centennially stable SOC fraction. Both variables are within the three most important predictor variables for $PARTY_{SOC}v2.0$

and $PARTY_{SOC}v2.0_{EU}$ while $TOC_{RE6}$ was ranked sixth or ninth out of the 18 predictor variables (Table 2). Other Rock-Eval® parameters related to the pyrolysable SOC fraction ($PC/TOC_{RE6}$ and HI, both negatively related to the centennially stable SOC proportion) were also important predictor variables for both models. The results suggest that a simple decreasing function of total SOC content cannot accurately predict the centennially stable SOC proportion in topsoils, according to the recent report by Clivot et al. (2019). They also confirm the generic elemental stoichiometry of the centennially stable SOC

fraction: it is consistently depleted in hydrogen (Barré et al., 2016; Gregorich et al., 2015; Poeplau et al., 2019); and they illustrate the usefulness of the pyrolysis step of Rock-Eval® thermal analysis and its volatile hydrocarbon effluents quantification to infer the proportion of the centennially stable SOC fraction in unknown topsoils.

### 4.2 Capability of the second version of $PARTY_{SOC}$ to partition the centennially stable and active SOC fractions

The learning set of the second version of the $PARTY_{SOC}$ statistical model was significantly diversified compared with the

first version. Its reference topsoil samples now represent wider pedoclimatic conditions (supplementary Table S2), and it includes one long-term vegetation change site as reference site (La Cabaña). Reference topsoils from the Colombian site of La Cabaña fit well into the global learning set of the statistical model: they did not alter its overall performance. The root-mean-square errors of $PARTY_{SOC}v2.0$ (internal or random splitting validation procedures) are comparable to the ones of the model's first version, where the content of the centennially stable SOC fraction was inferred exclusively from plant-free soils

(Fig. 2a, Table 3; Cécillon et al., 2018). Similarly, the expansion of the reference learning topsoil sample set to new soil types (Acrisol at La Cabaña, Chernozem at Bad Lauchstädt; FAO, 2014), soil texture (loamy coarse sand at Askov; supplementary Table S2), soil pH (in $H_2O$, with values as low as 4 at La Cabaña; supplementary Table S2) and climate (tropical at La Cabaña; supplementary Table S2) did not alter the performance of the model, when assessed using the internal or random splitting validation procedures (Fig. 2a, Table 3). Conversely, the leave-one-site-out validation procedure

illustrated that the second version of $PARTY_{SOC}$ is currently not capable of accurately partitioning SOC into its centennially



stable and active fractions in soil samples coming from pedoclimates that differ strongly from the ones included in the learning set (sites of La Cabaña and Bad Lauchstädt; Fig. 2b, Table 3b). This indicates that like all machine-learning approaches, the PARTY$_{SOC}$ model gains progressively more genericity (*i.e.*, capability to fairly predict the centennially stable SOC proportion in unknown soils) as its learning set integrates soils from new pedoclimates. To this respect, the

second version of PARTY$_{SOC}$ significantly extends the model's validity range to new pedoclimates (tropical Cambisols, continental Chernozems and temperate loamy coarse sand Luvisols). Contrarily, the relatively high prediction error of both PARTY$_{SOC}$v2.0 and PARTY$_{SOC}$v2.0$_{EU}$ models at Rothamsted (high $_R$RMSEP), a site with a pedoclimate rather similar to some of the other European sites included in the learning set of PARTY$_{SOC}$, may be due to an inaccurate estimate (overestimation) of the centennially stable SOC content at this site. Indeed, a report from an ancient LTBF trial at

Rothamsted (drain gauge experiment; Jenkinson and Coleman, 1994), on the same soil type than the Highfield bare fallow experiment, showed a measured total SOC content of 7.9 g C kg$^{-1}$, which is lower than our current estimate of the centennially stable SOC content (9.72 g C kg$^{-1}$; Table 1). Yet, the conditions of the drain gauge experiment, with a basic soil pH value of 7.9 due to heavy dressing of chalk on Rothamsted's arable lands before the 19[th] century (Avery and Catt, 1995; Jenkinson and Coleman, 1994), may not be directly comparable to the conditions of the Highfield bare fallow experiment

showing acidic pH values ranging from 5.2 to 6.3 (supplementary Table S2).

The predictive ability of the second version of PARTY$_{SOC}$ was more thoroughly assessed compared to the first version of the statistical model. Specifically, the sensitivity of model performance to the reference sites included in the learning set demonstrates that local models —with learning sets composed of soils from pedoclimates similar to the ones of the soils from the prediction set— showed better predictive ability of the centennially stable SOC proportion compared to a global

statistical model (Fig. 3). While the current learning set is composed of too few reference sites to implement local modelling, this suggests that the European version PARTY$_{SOC}$v2.0$_{EU}$ should be preferred to the global PARTY$_{SOC}$v2.0 model when predicting the centennially stable SOC proportion in unknown soils from Europe. The mean prediction error of 0.15 obtained using the leave-one-site-out validation procedure of PARTY$_{SOC}$v2.0$_{EU}$ (with a $_R$RMSEP of 0.27; Table 3a) is probably a

conservative estimate of the accuracy of this model to partition the centennially stable and active SOC fractions over a wide pedoclimatic range of agricultural topsoils in Northwestern Europe.

### 4.3 Future developments and recommended applications of the second version of the PARTY$_{SOC}$ model

The second version of the PARTY$_{SOC}$ model is based on six long-term agricultural sites including an LTBF treatment located in Northwestern Europe and one vegetation change (C$_4$ to C$_3$ plants) site located in Colombia. The very first future

improvement for the machine-learning model is to pursue the expansion of the pedoclimatic diversity of its learning set. A few additional LTBF sites and several C$_3$ to C$_4$ plants (or C$_4$ to C$_3$) long-term vegetation change sites (including space-for-time substitution, like the site of La Cabaña) could be used to achieve this goal. A potential complement lies in a few long-term experimental sites with soil archives and treatments experiencing contrasting SOC stock changes. Radiocarbon





measurements on recent and archived soil samples from such sites can be used to infer the content of the centennially stable
SOC fraction in topsoils (Hsieh, 1992), but also in subsoils, to allow extending the model to deeper soil horizons. Following
the method developed by Buyanovsky and Wagner (1998b, 1998a) and Huggins et al. (1998), the content of the centennially
stable SOC fraction can also be estimated at a few additional long-term experiments with contrasted SOC inputs. A
promising complement to these strategies lies in numerous long-term sites where time series of SOC inputs, outputs and
stocks are well constrained (*i.e.*, long-term experiments or long-term monitoring sites in various types of ecosystems
including arable land, grassland and forest). It is possible to reliably infer the content of the centennially stable SOC fraction
at these sites using simple models of SOC dynamics like AMG (Clivot et al., 2019). Combining all these strategies could
help expanding significantly the learning set of PARTY$_{SOC}$ to soil samples from diverse climates, ecosystems, soil types and
soil depths. When the learning set of PARTY$_{SOC}$ will integrate a sufficient diversity of soil samples, a second future
improvement of the model lies in the comparison of different machine-learning algorithms as well as the testing of local
modelling approaches, as commonly used in soil spectroscopy studies (Dangal et al., 2019; Gogé et al., 2012; Ramirez-
Lopez et al., 2013b, 2013a).

Meanwhile, the current version of the PARTY$_{SOC}$v2.0 model and especially its European version PARTY$_{SOC}$v2.0$_{EU}$ already
provide accurate predictions of the size of the centennially stable and active SOC fraction in agricultural topsoils of a large
diversity of pedoclimatic conditions (Fig. 2; Table 3). We consider that PARTY$_{SOC}$v2.0$_{EU}$ is mature enough (see Sect. 3.3,
3.4 and 4.2) to be reliably applied on agricultural topsoils in Northwestern Europe, or to be tested on topsoils of other
ecosystems under similar pedoclimates for research purposes. The PARTY$_{SOC}$v2.0$_{EU}$ model is available on public
repositories as an R script and an R data file (see Sect. Data and code availability). PARTY$_{SOC}$v2.0$_{EU}$ generates predictions
of the centennially stable and active SOC proportions and contents (in g C kg$^{-1}$; obtained by multiplying the centennially
stable and active SOC proportions by TOC$_{RE6}$) in unknown soil samples, using their measured Rock-Eval® parameters.

The second version of PARTY$_{SOC}$ enables the reliable partitioning of SOC into its centennially stable and active SOC
fractions (Fig. 2). The validation of the model at the scale of Northwestern Europe presented here (PARTY$_{SOC}$v2.0$_{EU}$)
constitutes a breakthrough in the metrology of SOC kinetic pools. It represents a great improvement compared to other
approaches that consistently fail to achieve a proper separation of active from stable SOC (Fig. 1; Hsieh, 1992; von Lützow
et al., 2007). Those methods such as the physical or physico-chemical SOC fractionation schemes have been developed to
initialize the size of SOC kinetic pools of models (Skjemstad et al., 2004; Zimmermann et al., 2007a) and some of them are
now implemented on large topsoil sample sets at the national or continental scale in Europe (Cotrufo et al., 2019; Vos et al.,
2018) and Australia (Gray et al., 2019; Viscarra Rossel et al., 2019). A similar implementation in soil monitoring networks
of Rock-Eval® measurements combined with the second version of PARTY$_{SOC}$ will provide a more accurate quantification
of the functionally different SOC fractions that are centennially stable or active (Fig. 1). Large-scale Rock-Eval®
measurements and the combined application of the PARTY$_{SOC}$v2.0$_{EU}$ model are already ongoing in the French soil



monitoring network for soil quality assessment (RMQS; Jolivet et al., 2018). We recommend undertaking similar works in other national and international soil monitoring networks. The second version of PARTY$_{SOC}$ can also be directly employed

as a SOC pools partitioning method for simple models of SOC dynamics that are built on the same dualistic conceptual approach of SOC persistence (*i.e.*, active *vs.* inert SOC pools). The accuracy of these simple models, such as AMG, is highly sensitive to the proper partitioning of SOC kinetic pools (Clivot et al., 2019), and could thus strongly benefit from the second version of PARTY$_{SOC}$.

We envision a significant contribution of the PARTY$_{SOC}$ machine-learning model based on Rock-Eval® thermal analysis to the forthcoming large-scale availability of accurate information on the size of the centennially stable or active SOC fractions. Such accurate information will foster (1) the initiatives of soil health assessment and monitoring and (2) the modelling works of SOC dynamics and of the climate regulation function of soils.

**Data and code availability**

The Rock-Eval® data of the 105 reference topsoil samples of PARTY$_{SOC}$v2.0 are provided in supplementary Table S3. The R script used to extract Rock-Eval® 6 raw data and calculate Rock-Eval® parameters; the Rock-Eval® data and the R script used to build PARTY$_{SOC}$v2.0 and PARTY$_{SOC}$v2.0$_{EU}$ models and test their performance; and the PARTY$_{SOC}$v2.0$_{EU}$ model (available as an R script and an R data file) can be accessed on GitHub at https://github.com/lauric-cecillon/PARTYsoc and on Zenodo at the permanent link https://doi.org/10.5281/zenodo.4446138.

**Acknowledgments**

The French Agence nationale de la recherche (StoreSoilC project, grant ANR-17-CE32-0005), the French Agence de la transition écologique (ADEME), and Ville de Paris (SOCUTE project, emergence(s) program) funded this research. We are indebted to the generations of technicians and scientists that started and managed the long-term experiments and archives of soil samples used in this work. We thank Rothamsted Research for access to samples and data from the Rothamsted Sample

Archive and the electronic Rothamsted Archive (e-RA). The Rothamsted Long-term Experiments are supported by the UK Biotechnology and Biological Sciences Research Council under the National Capabilities programme grant (BBS/E/C/000J0300), and by the Lawes Agricultural Trust. We thank David Montagne and Joël Michelin (AgroParisTech, France) who provided information on the soil characteristics at Grignon. We thank our colleagues of the Soil Science research group at Ecole normale supérieure (Paris, France), especially Samuel Abiven, Núria Catalán, Bertrand Guenet and

Marcus Schiedung who provided advices that improved this manuscript.



**Author contributions**

L.C. and P.B. designed the study with contributions from C.C. and F.B.. F.B. and F.S. performed the Rock-Eval® measurements. L.C. wrote the R scripts used to calculate Rock-Eval® parameters and built the second version of the PARTY$_{SOC}$ model with contributions from P.B., L.N.S. and E.K.. B.T.C., U.F., S.H., T.K., I.M., F.v.O, C.P., J.C.Q. provided the topsoil samples and the metadata of the reference sites. L.C. and P.B. wrote the manuscript with contributions from all authors.

**Competing interests**

The authors declare that they have no conflict of interest.

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
