# Peer review of "Partitioning soil organic carbon into its centennially stable and active fractions with statistical models based on Rock-Eval® thermal analysis (PARTYSOCv2.0 and PARTYSOCv2.0EU)"

_Geoscientific Model Development, 2021_

## Author Comment (AC2)

[Figure]

**Figure S1: Sensitivity of the PARTY$_{SOC}$v2.0$_{EU}$ machine-learning model to the test set. Model sensitivity to the test set was assessed as its sensitivity to independent test samples (1) from a reference soil group not existing in the training set (*i.e.* excluding Chernozem soil samples from the test set); (2) that are unlikely to be encountered in agricultural soils (*i.e.* excluding from the test set soils sampled at late dates of bare fallow treatments, more than 25 years after the experiment onset, which cannot represent soils with regular carbon input).** This figure is built with a test set composed exclusively of soils from independent Cambisols and Luvisols of Northwestern Europe. It should be compared to Fig. 2b (for PARTY$_{SOC}$v2.0$_{EU}$) that is built using the exact same training set and validation procedure (leave-one-site-out), but that uses a different test set including 32 additional independent test samples (15 Chernozem soils samples and 17 soil samples collected at late dates in bare fallow treatments).

---

## Author Response (AR1)

**Reply to reviewer 1 (Dr. E. Lugato)**

R1: The authors present an improved version of a method previously published (Biogeosciences, 15, 2835–2849, 2018), combining thermal analysis and machine learning in order to predict a centennial stable and active soil organic pool. It is certainly an interesting approach, that can provide useful indications for monitoring soil organic carbon and quality changes. The methodology is amply described as well as the validation process, therefore, I have only minor recommendations.

Reply: we deeply thank Dr. Lugato (reviewer 1) for his positive feedback on our work and manuscript (MS).

R1: Personally, I would be more careful in recommending the application of PARTYSOCv2.0EU in European agricultural topsoils (line 43), since the performances were lower both in the 'leave-one-out-site' and in the two independent sites validation. This is an indication that PARTYSOCv2.0EU would benefit of additional training sites; due to great variability of pedo-climatic and agroecosystem conditions in Europe, application of machine learning methods outside the range of their training can be critical.

Reply: we fully agree with Dr. Lugato and also with reviewer 2 (see our reply to reviewer 2) that the application of machine-learning models outside the range of their training set cannot be recommended.

We accordingly revised the main text, specifying that "*To this respect, we consider that applying the second version of PARTY$_{SOC}$ to unknown soils from pedoclimates outside its training set cannot be recommended*" (lines 607–608 of the revised MS). We also revised the abstract accordingly, removing our overoptimistic recommendation highlighted by Dr. Lugato and also by reviewer 2. We further refined the text of the abstract with the following sentence: "*More specifically, our results show that PARTY$_{SOC}$v2.0$_{EU}$ reliably partitions SOC kinetic fractions at its Northwestern European validation sites on Cambisols and Luvisols, which are the two dominant soil groups in this region*" (lines 41–42 of the revised MS). This refined statement is based on new results of the model (sensitivity analysis to the test set) presented at lines 527–532 of the revised MS, and in the new supplementary Figure S1.

R1: Also the discussion presents, sometime, some repetitive concepts.

Reply: we thank Dr. Lugato for this comment. We significantly revised the Discussion section of the MS. First we tried to avoid repetitions (we shortened the Sections 4.2 and 4.3, removing a full paragraph from Section 4.3); second we refined the discussion paragraphs related to the applicability of PARTY$_{SOC}$v2.0$_{EU}$ in European agricultural topsoils, as meaningfully recommended by both Dr. Lugato and reviewer 2 (see our reply to reviewer 2).

R1: I would also have expected more about comparison with other approaches especially in term of cost-benefit. While this method can be applied on existing monitoring schemes (as it requires a soil sample), there is no information of the cost of the thermal analysis, the complexity etc., which are interesting aspects if the aim is to propose a routine method.

Reply: we thank Dr. Lugato for his meaningful suggestion. We added in the main text some information regarding the cost of Rock-Eval® thermal analysis: "*These characteristics are measurable quickly (ca. 1 h per sample) and at a reasonable cost (less than 60 USD per*

*sample in private laboratories) using Rock-Eval® thermal analysis*" at lines 100–102 of the revised MS.

R1: Line1: I would suggest adding in brackets after 'active' (with turnover time of months to a few years).

Reply: In our work, we are referring to a centennially active soil organic carbon (SOC) fraction, which means that it has a typical mean turnover rate of a few decades. To clarify this point, we modified the corresponding sentence in the abstract "*stable or active on a century scale*" (line 28 of the revised MS). Information related to this point is also provided in the main text: "*The most drastic conceptual simplification of SOC persistence considers only two pools: (1) one made of young SOC with a short turnover rate (typically three decades on average; the active SOC pool) and (2) one made of older SOC that persists much longer in the soil (more than a century; the stable, passive or persistent SOC pool)*" (lines 61–64 of the revised MS).

R1: Line 77-80: it seems that the approach proposed is quite insensitive to the number of samples training the model. Indeed only 7 sites are used and the 'leave-one-site-out' validation is worse that the 'internal'. This is a likely sign that also this model would benefit from additional training. So I don't understand the concept that other methods 'need to be inferred from statistical models or infrared spectroscopy'. Isn't it also what the authors are doing, training a ML on measurements?

Reply: we agree with Dr. Lugato that the PARTY$_{SOC}$ model would benefit from additional training sites, as already stated in the discussion section, see *e.g.* "*The very first future improvements to the PARTY$_{SOC}$ machine-learning model are to increase the size and further expand the pedoclimatic diversity of its training set*", at lines 625–626 of the revised MS.

At lines 76–81 of the revised MS, we are actually addressing a different point, *i.e.* that only a few techniques are "*reasonably reproducible and easy to implement such as the ones based on rapid thermal analysis and chemical extractions*" compared to other techniques such as the tedious "*size and density SOC fractionation*" methods. On the other hand, we recognize that the technical difficulties in implementing size and density SOC fractionation methods on large soil sample set can be resolved by inferring their results from regression models based on environmental variables (*e.g.* Cotrufo et al., 2019) or infrared spectroscopy (*e.g.* Viscarra Rossel et al., 2019).

We would like to point out that such works are using machine-learning in a way that differs from our approach. Our approach uses machine-learning to link Rock-Eval® parameters to long-term observations of soil carbon persistence, but it always requires the use of a Rock-Eval® measurement to infer soil carbon pool partitioning in a new soil sample. This is not the case for the approaches of *e.g.* Viscarra Rossel et al. (2019) or Cotrufo et al. (2019) that use machine-learning with the only goal of avoiding the use of tedious physico-chemical soil carbon fractionation methods on numerous soil samples. These approaches then predict, using only infrared spectroscopy or environmental information that are available from new soil samples or new sites, the size of different soil carbon fractions corresponding to a mixture of centennially stable and active carbon, as illustrated in our conceptual Figure 1.

R1: Line 82. I generally agree, although I would like to point out that some fractionation methods don't necessary aim at isolating kinetically defined pools, but rather pools underlying pathways of SOC formation and stabilization (e.g. the work on MEMS by Cotrufo et al.).

Reply: we agree with Dr. Lugato that some fractionation methods may aim at isolating pools with specific formation, stabilization and destabilization process, but we first note that most of these attempts have proven to be unsuccessful (von Lützow et al., 2007). Second, when it comes to modeling soil carbon dynamics, isolated soil carbon fractions are usually transformed to conceptual kinetic pools with a unique decay rate, which is not correct since the isolated fractions are most often a mixture of fast-cycling and slow-cycling carbon (von Lützow et al., 2007). This is the case of the MEMS model that is "*directly equating mineral-associated organic matter and particulate organic matter fractions with corresponding model pools*" (Robertson et al., 2019), assigning a unique decay rate for the particulate or the mineral-associated fractions.

R1: Lines 85-86: yes, unless models are built to predict model fractions (eg. MEMS, COMISSION and others)

Reply: we agree with Dr. Lugato that models are meaningful to test competing theories on soil carbon dynamics. On the other hand, it seems obvious from the literature that models of SOC dynamics such as the MEMS model are not built for the sole purpose of predicting model fractions. Clearly, the MEMS model was built to forecast soil organic carbon dynamics "*it does provide the basis for a more mechanistic model that can simulate SOM dynamics on the ecosystem scale*" (Robertson et al., 2019).

R1: Fig.1 left panel. Is it a conceptual figure or size fractions were effectively analyzed by RockEval?

Reply: Figure 1 is a conceptual figure illustrating that all soil carbon fractionation methods (*e.g.* size-based methods, thermal methods, *etc.*) are consistently isolating fractions that are a mixture of centennially stable and active carbon. In this Figure, we state that is only when calibrated on long-term observational data of SOC persistence (as done by the PARTY$_{SOC}$ machine-learning model) that fractionation methods such as Rock-Eval® thermal analysis can reliably quantify the respective sizes of the centennially stable and active SOC fractions. For information, some Rock-Eval® measurements of soil carbon size or chemical fractions are reported in previously published papers (Saenger et al., 2015; Soucémarianadin et al., 2019; Poeplau et al., 2019).

R1: Line 195: It seems that authors suggest that the lowest SOC treatment received less 'unwanted' C input, while its lower value may be due to any source of uncontrolled variability. Are the results very sensitive to this approach and is there any risk to, opposite, underestimate the centennial carbon pool?

Reply: we thank Dr. Lugato for this comment. We agree that a value of total SOC content lower than the site-specific estimate of the centennially stable soil carbon content may indicate an uncontrolled within-site variability of the soil carbon content (and of the centennially stable carbon content). However, we consider that the risk of overestimating the centennially stable soil carbon content using the curve-fitting method proposed by Barré et al. (2010) is much higher (as stated at lines 190–196 of the revised MS) than the risk of underestimating it using the modified methodology proposed in our MS. This is best

illustrated at the site of Rothamsted (England), where the lowest total SOC measurement (9.72 gC/kg) is lower than our curve-fitting estimate of the centennially stable carbon content at that site (10.46 gC/kg; Cécillon et al., 2018), while this value (retained as the more accurate site-specific value of centennially stable soil carbon content, see Table 1) remains much higher than another estimate of the centennially stable soil carbon content (7.9 gC/kg) reported for another bare fallow experiment at the same site and for the same soil type by Jenkinson and Coleman (1994; see the Discussion section at lines 614–623 of the revised MS).

R1: Line 275: maybe 'inferred' is better than 'calculated' as a fitting procedure was generally used.

Reply: "calculated" was replaced with "inferred" at lines 274 and 276 of the revised MS.

R1: Table 1: while the Centennial stable SOC is a unique value per site, what does the SOC content refer to (average over all treatments and years within a site)?

Reply: we thank Dr. Lugato for this comment; the basic statistics of SOC content reported in Table 1 refer to the reference soil samples used (15 per reference site). We clarified this point in the revised version of Table 1.

R1: Line 358: if I well understood, out of those 105 samples, the centennial stable pool was inferred only from the LTBF and, then, assumed to be the same for all other treatments within the same site. I was wandering if some agronomic pracrices (for instance organic application) can bias this assumption. In fact, as far as understood (line 315), treatments with repeated application of some types of exogenous organic matter were not considered. My question is whether this poses a limit in the wide applicability of the method, since lot of soils receive manure and compost in Europe.

Reply: we thank Dr. Lugato for his comment and question regarding the influence of exogenous organic matter (EOM) applications to soil on its centennially stable carbon content.

While this is beyond the scope of the present work, we agree that the question raised by Dr. Lugato is important. As stated at lines 312–320 of the revised MS, we suspect that repeated applications of EOM to soil may increase the content of the centennially stable SOC fraction after several decades (*e.g.* biochar).

However, we would like to point out that such effects of repeated EOM applications on the centennially stable soil carbon content only impacts the selection of reference sites for the PARTY$_{SOC}$ machine-learning model (as explained in details at lines 312–320 of the revised MS), but that these effects do not affect the applicability of our method on unknown soils (as long as the PARTY$_{SOC}$ machine-learning model is able to capture these effects).

R1: Table 2. Adding one site (La Cabana), the rank of the variable importance changed as well as the predicted centennial SOC proportion on a different extent depending on sites (Fig. 2b, Fig. 3a,b). Have authors considered to introduce additional variables in the Random Forest model (eg. texture) to make it more robust?

Reply: we thank Dr. Lugato for this meaningful comment. The current version of the PARTY$_{SOC}$ machine-learning models is based on a limited number of reference sites and is thus, in its current configuration, sensitive to the addition of new sites, as illustrated by the results of the sensitivity analysis to the training set, presented in Figure 3 and in the supplementary Table S5.

We agree with Dr. Lugato that adding predicting variables, such as soil texture, could potentially improve the performance and the robustness of the PARTY$_{SOC}$ machine-learning model. However, as stated above, the model is currently based on a limited number of reference sites, and we think that adding soil and environmental properties as predicting variables may, on the other hand, reduce the robustness of the model when applied to new sites. This option is nevertheless under consideration for the next version of the model that will integrate a significant number of additional reference sites in its training set.

**References**

Barré, P., Eglin, T., Christensen, B. T., Ciais, P., Houot, S., Kätterer, T., van Oort, F., Peylin, P., Poulton, P. R., Romanenkov, V., and Chenu, C.: Quantifying and isolating stable soil organic carbon using long-term bare fallow experiments, Biogeosciences, 7, 3839–3850, https://doi.org/10.5194/bg-7-3839-2010, 2010.

Cécillon, L., Baudin, F., Chenu, C., Houot, S., Jolivet, R., Kätterer, T., Lutfalla, S., Macdonald, A., van Oort, F., Plante, A. F., Savignac, F., Soucémarianadin, L. N., and Barré, P.: A model based on Rock-Eval thermal analysis to quantify the size of the centennially persistent organic carbon pool in temperate soils, Biogeosciences, 15, 2835–2849, https://doi.org/10.5194/bg-15-2835-2018, 2018.

Cotrufo, M. F., Ranalli, M. G., Haddix, M. L., Six, J., and Lugato, E.: Soil carbon storage informed by particulate and mineral-associated organic matter, Nat. Geosci., 12, 989–994, https://doi.org/10.1038/s41561-019-0484-6, 2019.

Jenkinson, D. S. and Coleman, K.: Calculating the annual input of organic matter to soil from measurements of total organic carbon and radiocarbon, Eur J Soil Science, 45, 167–174, https://doi.org/10.1111/j.1365-2389.1994.tb00498.x, 1994.

von Lützow, M., Kögel-Knabner, I., Ekschmitt, K., Flessa, H., Guggenberger, G., Matzner, E., and Marschner, B.: SOM fractionation methods: Relevance to functional pools and to stabilization mechanisms, Soil Biology and Biochemistry, 39, 2183–2207, https://doi.org/10.1016/j.soilbio.2007.03.007, 2007.

Poeplau, C., Barré, P., Cécillon, L., Baudin, F., and Sigurdsson, B. D.: Changes in the Rock-Eval signature of soil organic carbon upon extreme soil warming and chemical oxidation - A comparison, Geoderma, 337, 181–190, https://doi.org/10.1016/j.geoderma.2018.09.025, 2019.

Robertson, A. D., Paustian, K., Ogle, S., Wallenstein, M. D., Lugato, E., and Cotrufo, M. F.: Unifying soil organic matter formation and persistence frameworks: the MEMS model, Biogeosciences, 16, 1225–1248, https://doi.org/10.5194/bg-16-1225-2019, 2019.

Saenger, A., Cécillon, L., Poulenard, J., Bureau, F., De Daniéli, S., Gonzalez, J.-M., and Brun, J.-J.: Surveying the carbon pools of mountain soils: A comparison of physical fractionation and Rock-Eval pyrolysis, Geoderma, 241–242, 279–288, https://doi.org/10.1016/j.geoderma.2014.12.001, 2015.

Soucémarianadin, L., Cécillon, L., Chenu, C., Baudin, F., Nicolas, M., Girardin, C., Delahaie, A., and Barré, P.: Heterogeneity of the chemical composition and thermal stability of particulate organic matter in French forest soils, Geoderma, 342, 65–74, https://doi.org/10.1016/j.geoderma.2019.02.008, 2019.

Viscarra Rossel, R. A., Lee, J., Behrens, T., Luo, Z., Baldock, J., and Richards, A.: Continental-scale soil carbon composition and vulnerability modulated by regional environmental controls, Nat. Geosci., 12, 547–552, https://doi.org/10.1038/s41561-019-0373-z, 2019.

**Reply to reviewer 2**

R2: This work presents a model (updated from previously published work) that predicts the proportion of centennially stable soil organic carbon (SOC). The manuscript is well written, the method is interesting, and described in a good level of detail. However, I have raised a number of concerns in the text below which need to be addressed. My primary concern being how accurate the model really is, given the poor performance when applied to sites outside the training dataset. Since the performance is so poor, I think the conclusions are overly positive and I question the recommendation that this model be used for other sites outside of the training dataset.

Reply: we deeply thank reviewer 2 for spending some of his/her time to evaluate our work.

In the lines below, we discuss the main concern of reviewer 2 regarding the actual performances of the second version of the PARTY$_{SOC}$ machine-learning model, *i.e.* its accuracy to predict the respective sizes of the centennially stable and active soil organic carbon (SOC) fractions in unknown soils.

In particular, we point out that the model performance is not "so poor", as stated by reviewer 2, notably for a large portion of Cambisols and Luvisols, two major soil types in Europe (European Commission, 2008), where soil organic pool partitioning is currently unknown and urgently needed to improve our understanding and projections of the fate of soil carbon. Our assertion is supported by new results (added in the Results section at lines 527–532 of the revised MS, and reported in the new supplementary Figure S1). The text has been changed accordingly (see the revised abstract and discussion on this specific topic).

On the other hand, we fully agree with reviewer 2 and also with reviewer 1 (see our reply to reviewer 1) that the application of machine-learning models outside the range of their training set cannot be recommended. We accordingly revised the abstract and the main text, specifying for instance that "*To this respect, we consider that applying the second version of PARTY$_{SOC}$ to unknown soils from pedoclimates outside its training set cannot be recommended*" (lines 607–608 of the revised MS).

We also followed the recommendations of both reviewer 1 and 2 regarding the tone of our conclusions that appeared to be overly optimistic in the abstract and in the main text. We significantly toned down our conclusions in the revised MS and we further refined them in link with the new results mentioned above, regarding the applicability of the model to unknown European agricultural soils.

R2: Specific comments #1: Inconsistent handling of sites

Reply: we disagree with reviewer 2 and we detail below how reference sites, soil samples and their carbon analysis were, for this second model version, handled more rigorously compared with the first published version of the PARTY$_{SOC}$ machine-learning model (Cécillon et al., 2018).

R2: Table 1 and the text outlines that different ways of measuring SOC content and different ways of calculating the centennially stable SOC fraction content were used for different sites. This is concerning as it is not clear what is being predicted here.

Reply: at all sites, SOC contents were determined by elemental analysis (dry combustion) and by Rock-Eval® thermal analysis. Here, we have chosen to use SOC contents determined by Rock-Eval® to determine the centennially stable soil carbon proportions in reference soil samples, contrary to the first version of the model that used SOC contents determined by elemental analysis (this modification of the PARTY$_{SOC}$ model is listed in Table S1 and described in the Methods section of the MS).

As explained in the main text (lines 275–277 of the revised MS): "*the reason behind this choice was to link the Rock-Eval® parameters measured on a reference topsoil sample to a calculated proportion of the centennially stable SOC fraction that better reflected the organic carbon that actually evolved during its Rock-Eval® analysis*".

The SOC contents determined by Rock-Eval® were either obtained as the TOC$_{RE6}$ parameter (5 reference sites) or alternatively as the sum of TOC$_{RE6}$+MinC parameters (reference sites of Askov and Bad Lauchstädt; Table 1). The rationale behind the selection of TOC$_{RE6}$ or TOC$_{RE6}$+MinC for determining SOC content is straightforward and already described in a specific paragraph of the Methods section (lines 273–294 of the revised MS). Briefly, TOC$_{RE6}$ was used as a proxy of SOC content at sites where TOC$_{RE6}$ reliably reproduced the value determined by elemental analysis without any systematic bias (5 sites out of 7). At the reference sites of Askov and Bad Lauchstädt, however, TOC$_{RE6}$ parameter systematically underestimated the results from elemental carbon analysis, while adding the MinC parameter to TOC$_{RE6}$ improved the carbon yield of Rock-Eval® analysis. The sum of TOC$_{RE6}$+MinC was thus used to determine SOC content of the 30 reference soil samples from Askov and Bad Lauchstädt.

R2: This is especially true since Table S2 and Table 1 show very different values of centennially stable SOC fraction calculated from the Bayesian curve-fitting and that measured on site. I'm not convinced there is a valid reason to ever have inconsistent handling of sites, and if there is a reason, it needs to be thoroughly explained in the text.

Reply: we disagree with reviewer 2 on this point. We argue that the values of the centennially stable SOC contents calculated from the Bayesian curve-fitting method and those measured on-site are not differing strongly (5.50 gC/kg *vs.* 6.22 gC/kg at Versailles, 9.72 gC/kg *vs.* 10.46 gC/kg at Rothamsted, 15 gC/kg *vs.* 16.22 gC/kg at Bad Lauschstädt, 4.75 gC/kg *vs.* 5.12 gC/kg at La Cabaña; see Tables 1 and S2). In particular, the hierarchy in the centennially stable SOC content of the seven reference sites is unchanged whatever the calculation method.

The reason for preferring on-site measurements to the Bayesian curve-fitting method at four out of the seven reference sites is described in the main text. At lines 190–196 of the revised MS, we explain that this choice has been made to reduce the risk of overestimating the actual value of the centennially stable SOC content, as SOC inputs to bare fallow topsoils are low but not null, and as C$_4$-plant derived SOC inputs are possible after conversion to C$_3$ plants (Jenkinson and Coleman, 1994; Petersen et al., 2005). This modification (improvement) of the PARTY$_{SOC}$ machine-learning model, compared to the first published version, is also listed in Table S1.

On the other hand, we agree with reviewer 2 that this point could be better explained in the main text. We thus added a few sentences on this topic in the Results section (lines 432–437 of the revised MS): "*These values did not differ strongly from the values of the centennially*

*stable SOC contents calculated from the Bayesian curve-fitting method (Table 1, Table S2). In particular, the hierarchy in the centennially stable SOC content of the seven reference sites was unchanged whatever the calculation method. These values were retained as the best estimates of the site-specific content of the centennially stable SOC fraction in topsoils of the four sites in order to reduce the risk of overestimating the actual value of the centennially stable SOC content, compared to the first published version of the model (see Sect. 2.1, Table 1, supplementary Table S1)*".

R2: Specific comments #2: Needs discussion on which validation method is more informative

R2: Two methods of validation are presented in the text and visually shown in Figure 2. These two methods show substantially different accuracies of the method. Whilst I think it is helpful to have both methods included, I think discussion on which method should be used to judge the model needs to be included. For example, from my understanding of the method, the test dataset for "internal validation" is made up of data from the sites in the training dataset. This means the test dataset is not truly independent of the training dataset, so it is not surprising that the "internal validation" looks a lot better.

Reply: we thank reviewer 2 for this comment. In fact, as described in the Methods section (lines 372–387 of the revised MS), three complementary procedures were used to assess the predictive ability of the PARTY$_{SOC}$ machine-learning model: (1) internal, (2) random splitting (mean statistics of 15 different models) and (3) leave-one-site-out validation procedures. To better clarify this point in the main text, we modified a sentence in the Methods section (lines 372–373 of the revised MS): "*The performance of PARTY$_{SOC}$v2.0 and PARTY$_{SOC}$v2.0$_{EU}$ was assessed by statistical metrics comparing the predicted vs. the estimated values of their reference topsoil sample set using three complementary validation procedures*".

Each validation procedure gives different information regarding the performance of the machine-learning model. Clearly, we agree with reviewer 2 that the leave-one-site-out validation procedure gives the most conservative estimation of the true uncertainty of regression models on unknown topsoil samples. We thus added a sentence for clarification (lines 386–387 of the revised MS): "*The "leave-one-site-out" validation should be seen as the procedure giving the most accurate estimation of the uncertainty of both regression models on unknown topsoil samples*" (see also changes made in the Discussion section at line 602 of the revised MS).

On the other hand, by definition, the leave-one-site-out validation procedure cannot help understanding the potential of the machine-learning model to reliably predict the centennially stable SOC proportion in soils from "extreme" pedoclimates that are present at only one reference site. This potential is better reflected by the internal and the random splitting validation procedures. That is to say that the leave-one-site-out procedure reveals that the model cannot predict the centennially stable SOC fraction in Chernozem soils (Bad Lauchstädt; see Figure 2b) or under tropical climate (La Cabaña; see Figure 2b) when no site with such pedoclimate is included in the training set. Conversely, the internal and the random splitting validation procedures suggest that once soils from Chernozem soil type or tropical climate are included in the training set, the predictions of the machine-learning model get more accurate for those soils (see Figure 2a; though this is not demonstrated in unknown Chernozem or tropical soils and will thus require further validation in the next version of the PARTY$_{SOC}$ model).

R2: Additionally, in the methods section (line 337), it is described how the training dataset was chosen to maximise the range of centennially stable SOC. This selection means that the testing dataset will likely have a more central range of values than the true range of values, potentially imparting a bias on the validation (i.e. there will be few extreme values in the test dataset).

Reply: we disagree with reviewer 2 on this point and we argue that his/her interpretation is not correct. As stated at lines 334–335 of the revised MS, "*the 15 topsoil samples retained for each reference site were selected: (1) to have a range of proportion of centennially stable SOC fraction as wide as possible*". This selection of reference soil samples at each site does not imply that the test sets have necessarily a more central range of values than the training sets. For the three complementary validation procedures (internal, random splitting, and leave-one-site-out; see above), soil samples with "extreme" values of centennially stable SOC proportion still have a reasonable probability to be included in the different test sets. The validation statistics are thus by no means "positively" biased by our selection of training and test samples.

On the contrary, the validation statistics presented in Table 3 could potentially be "negatively" biased by the presence, in many of the different test sets, of soils sampled at very late dates of bare fallow treatments (with extremely high proportions of centennially stable SOC, often above 0.8). For these "extreme" soil samples in the test sets, the machine-learning model often has difficulties to predict their centennially stable SOC proportion, whatever the validation procedure (Figure 2a, b). These "extreme" centennially SOC proportions, however, are very unlikely to occur in unknown topsoils that are not under long-term bare fallow treatments. To better reflect this, we added a sensitivity analysis to the test set, and reported these new results in Section 3.4 (see also below our reply to the specific comments #3 of reviewer 2, and the new material reported at lines 527–532 of the revised MS, and in the new supplementary Figure S1).

R2: Furthermore, when considering the application of this model, it would be applied to new sites not within the training dataset, as in the "leave-one-site-out validation". For these reasons, I do not believe that the "internal validation" method is the best way to judge this work, instead the "leave-one-site-out validation" is more useful when considering the accuracy of the method.

Reply: we have never stated in the abstract or in the main text of our MS that the internal validation procedure is the best way to judge our work. As described above, we have used three complementary procedures (internal, random splitting, and leave-one-site-out) to assess the predictive ability the PARTY$_{SOC}$ machine-learning model.

As also stated above, we agree with reviewer 2 that the leave-one-site-out validation procedure gives the most conservative estimation of the true uncertainty of regression models on unknown topsoil samples. We thus added a sentence to clarify this point (lines 386–387 of the revised MS): "*The "leave-one-site-out" validation should be seen as the procedure giving the most accurate estimation of the uncertainty of both regression models on unknown topsoil samples*" (see also changes made in the Discussion section at line 602 of the revised MS).

R2: Specific comments #3: How useful is this model?

R2: My next point continues from the previous point on validation. If only the "leave-one-site-out validation" is considered, the accuracy of the method is significantly reduced. Looking at Figure 2b and Table 3, I am not convinced that this model performs much better than if the mean of the training dataset (0.52 or 0.55 from Table 1) was predicted for every datapoint. For example, in Figure 2b, it appears a horizontal line y=0.52 or y=0.55 would equally predict the SOC. If this is true, then there is no point in applying the model to sites outside of the training dataset, so this needs to be addressed in the manuscript.

Reply: we thank reviewer 2 for this comment. The fact that the performances of the $PARTY_{SOC}$ machine-learning model are significantly reduced when assessed using the leave-one-site-out validation procedure was already clearly mentioned in the Results section, at lines 487–488 of the revised MS "*The predictive ability of both models decreased when assessed using the "leave-one-site-out" procedure (see Sect. 2.5; Fig. 2b)*". It was also already mentioned in the Discussion section (lines 602–605 of the revised MS).

On the other hand, while we agree that assessing visually (on Figure 2) the predictive ability of the $PARTY_{SOC}$ model (as done by reviewer 2) is useful, we argue that statistical metrics provide a more objective assessment of regression model performance. In particular, the ratio of performance to interquartile range (RPIQ, see Bellon-Maurel et al., 2010 for a detailed description of this indicator), with a value above 2 for the $PARTY_{SOC}v2.0_{EU}$ model (2.39, leave-one-site-out validation procedure; see Table 3), demonstrates that this model "performs better than if the mean of the training dataset was predicted for every datapoint", as questioned by reviewer 2. We thus argue that the statement of reviewer 2 "it appears a horizontal line y=0.52 or y=0.55 would equally predict the SOC" is not correct, and that on the contrary, it is useful to apply the $PARTY_{SOC}v2.0_{EU}$ model to unknown soils from pedoclimates similar to the ones of the training set.

We agree with reviewer 2, however, that the $PARTY_{SOC}v2.0_{EU}$ model cannot provide accurate predictions in all soil types from Europe. To refine our analysis of model performance, we conducted a sensitivity analysis to the test set, assessing more specifically the predictive ability of $PARTY_{SOC}v2.0_{EU}$ in 5 sites developed on two main soil types: Cambisols and Luvisols (two major soil types in Europe; European Commission, 2008). The performance statistics of the model (leave-one-site-out validation procedure) are improved when considering only these 5 sites.

We reported this new result in the main text (Results section 3.4, at lines 527–532 of the revised MS and in the new supplementary Figure S1): "*The sensitivity analysis to the test set showed that when excluding Chernozem samples from the test set (i.e., validating the model exclusively on independent samples from Cambisols or Luvisols), the performance statistics of $PARTY_{SOC}v2.0_{EU}$ were improved (leave-one-site-out validation procedure: $R^2$ of 0.56; RMSEP of 0.13; n = 75). The further removal of independent test soils that are unlikely to be encountered in agricultural Cambisols and Luvisols (soils sampled at late dates of bare fallow treatments, more than 25 years after the experiment onset) also improved the performance statistics of $PARTY_{SOC}v2.0_{EU}$ (supplementary Fig. S1; leave-one-site-out validation procedure: $R^2$ of 0.71; RMSEP of 0.11; n = 58)*".

R2: Specific comments #4: Overstatement of conclusions

R2: Given my previous comments on the validation and accuracy of the model, I think that the conclusions have been overstated: e.g. line 633 "the current version of the

PARTYSOCv2.0 model … already provide **accurate** predictions…", line 641 "…enables the **reliable** partitioning of SOC into its centennially stable and active SOC fractions", line 642 "The validation of the model … here (PARTYSOCv2.0EU) constitutes a **breakthrough** in the metrology of SOC kinetic pools. It represents a **great improvement**…". I have put examples of the words I do not think are justified in bold, given the points I have raised above.

Reply: We agree with reviewer 2 and also with reviewer 1 that some of our conclusions could have been considered as overoptimistic, both in the abstract and in the Discussion section. We significantly toned them down, and adapted them to the new results provided at lines 527–532 of the revised MS and in the new supplementary Figure S1 (see above).

Specifically, we removed the last sentence of the abstract (""). We added a new sentence in the abstract that reflects more closely the new results: "*More specifically, our results show that PARTY$_{SOC}$v2.0$_{EU}$ reliably partitions SOC kinetic fractions at its Northwestern European validation sites on Cambisols and Luvisols, which are the two dominant soil groups in this region*" (lines 41–42 of the revised MS).

The Discussion section was thoroughly revised accordingly (see Sections 4.2 and 4.3). In the Section 4.2, we modified two important paragraphs:

At lines 602–605 of the revised MS: "*On the other hand, the leave-one-site-out validation procedure, the most robust validation procedure (see Sect. 2.5), demonstrates that the second version of PARTY$_{SOC}$ is currently not capable of accurately partitioning SOC into its centennially stable and active fractions in soil samples coming from pedoclimates that differ strongly from the ones included in the training set (Fig. 2b, Table 3b)*".

At lines 607–614 of the revised MS: "*To this respect, we consider that applying the second version of PARTY$_{SOC}$ to unknown soils from pedoclimates outside its training set cannot be recommended. The sensitivity analysis to the test set, however, shows that PARTY$_{SOC}$v2.0$_{EU}$ reliably partitions SOC kinetic fractions at its validation sites on Cambisols and Luvisols (with a mean prediction error on the centennially stable SOC proportion of 0.11; see Sect. 3.4 and supplementary Fig. S1). Cambisols and Luvisols are the two dominant reference soil groups in Europe, covering more than 41% of European land areas (European Commission, 2008). Though the model test set does not include all the within-group pedological variability of Cambisols and Luvisols (FAO, 2014), this suggests that PARTY$_{SOC}$v2.0$_{EU}$ can accurately partition SOC into its centennially stable and active fractions on a significant portion of Northwestern European agricultural soils*".

We renamed Section 4.3 "*Future developments of the PARTY$_{SOC}$ model*" (line 624 of the revised MS). In the Discussion section 4.3, we removed the paragraph "~~Meanwhile, the current version of the PARTY$_{SOC}$v2.0 model and especially its European version PARTY$_{SOC}$v2.0$_{EU}$ already provide accurate predictions of the size of the centennially stable and active SOC fraction in agricultural topsoils of a large diversity of pedoclimatic conditions (Fig. 2; Table 3). We consider that PARTY$_{SOC}$v2.0$_{EU}$ is mature enough (see Sect. 3.3, 3.4 and 4.2) to be reliably applied on agricultural topsoils in Northwestern Europe, or to be tested on topsoils of other ecosystems under similar pedoclimates for research purposes~~"

that contains one of the conclusions meaningfully criticized by reviewer 2. Similarly, we removed the sentence "$_{SOC}$ ".

Finally, following the recommendations of reviewer 2, "*breakthrough*" was replaced with "*significant progress*" (line 644 of the revised MS); and "*great improvement*" was changed to "*improvement*" (line 645 of the revised MS).

R2: Technical comments:

R2: Line 29: "a more accurate modelling" does not quite make sense, it should be something like "a more accurate model" or "a more accurate understanding"

Reply: "a more accurate modelling" was replaced with "for more accurate models of the carbon cycle" (lines 29 of the revised MS).

R2: Table 2 caption: "Symbols for p-values: *** $p < 0.001$; ** $p < 0.01$; * $p < 0.05$; NS $p > 0.05$ = not significant" But these symbols are not used in this table?

Reply: this sentence was removed from the revised Table 2 caption.

R2: Line 318: "Therefore, to increase the likelihood of verifying our hypothesis…" This is a bit of an odd phrasing, sounds a bit like cherry-picking data to get the desired answer, though I appreciate that is not what is going on. I suggest rephrasing this.

Reply: the corresponding paragraph was rephrased (lines 311–320 of the revised MS).

R2: Line 499: "Conversely, the two models show only two Rock Eval® parameters in common out of their five most important ones…" Could this demonstrate that some of those parameters are actually not adding to the predictive power of the model? What happens to the accuracy of the model if just the top x parameters are used? Perhaps the model is overfitted to the training data?

Reply: we agree with reviewer 2 that the different ranking of Rock-Eval® predictor variables for PARTY$_{SOC}$v2.0 and PARTY$_{SOC}$v2.0$_{EU}$ regarding their respective importance in model predictions (except the most important variable; see Table 2) may indicate, not demonstrate, that some of the Rock-Eval® parameters are not adding much to the predictive power of the model. This would be logical as many Rock-Eval® parameters are correlated to each other. On the other hand, it also illustrates the sensitivity of the model to the addition of a new reference site (La Cabaña).

We disagree with reviewer 2, however, when he/she states that this could indicate that the model is overfitted to the training data. Contrary to other multivariate regression methods (*e.g.* partial least squares regression), random forests regression is, by mathematical construction of its algorithm, relatively insensitive to poorly informative predictor variables or to auto-correlation of predictor variables; see *e.g.* Strobl et al. (2009) for a presentation of the random forests algorithm, and for a discussion on the risk of overfitting in random forests modelling. Poorly informative predictor variables get a small variable importance in the model predictions, leaving the performance statistics of the model almost unaffected (to illustrate this, as listed in Table S1, the first version of the model used 30 Rock-Eval®

predictor variables compared to 18 for the second version, with comparable performance statistics for both versions).

R2: Line 576: "Reference topsoils from the Colombian site of La Cabaña fit well into the global learning set of the statistical model: they did not alter its overall performance." Not sure that is true? As stated in line 468: "For most of the calculated statistics, the European version of the model PARTYSOCv2.0EU showed better performances than the PARTYSOCv2.0 model"

Reply: in this sentence of the Discussion section, we are in fact not comparing PARTY$_{SOC}$v2.0 and PARTY$_{SOC}$v2.0$_{EU}$. We are comparing the predictive ability of PARTY$_{SOC}$v2.0 with the one of the first version of the model. To clarify this point, we associated the two corresponding sentences (lines 587–590 of the revised MS): "*Reference topsoils from the Colombian site of La Cabaña fitted well into the training set of the global model: they did not alter its overall performance, as the root-mean-square errors of PARTY$_{SOC}$v2.0 (internal or random splitting validation procedures) are comparable to the ones of the model's first version, where the content of the centennially stable SOC fraction was inferred exclusively from plant-free soils (Fig. 2a, Table 3; Cécillon et al., 2018)*".

R2: Line 585: "currently not capable of accurately partitioning SOC… in soil samples coming from pedoclimates that differ strongly from the ones included in the learning set…" This relates to my earlier points - I would argue that this is true for any sites not included in the training dataset given the poor performance of the "leave-one-site-out validation".

Reply: following the recommendations of both reviewer 1 and 2, we have thoroughly revised the Discussion section regarding this topic. Details of changes in the Discussion section are provided above, in our reply to the specific comments #4 of reviewer 2.

R2: Line 610 (and abstract): "…conservative estimate of the accuracy of this model…" Why is this a conservative estimate? It does not seem so, given the issues raised above.

Reply: as stated above, we have thoroughly revised the Discussion section regarding this topic, following the recommendations of both reviewer 1 and 2. More specifically, "conservative" was removed from the abstract and the main text.

R2: Table S3 is provided as an excel spreadsheet – a more generic file format would be better, such as csv or netcdf.

Reply: Table S3 is now provided as a csv file.

**References**

Bellon-Maurel, V., Fernandez-Ahumada, E., Palagos, B., Roger, J.-M., and McBratney, A.: Critical review of chemometric indicators commonly used for assessing the quality of the prediction of soil attributes by NIR spectroscopy, TrAC Trends in Analytical Chemistry, 29, 1073–1081, https://doi.org/10.1016/j.trac.2010.05.006, 2010.

Cécillon, L., Baudin, F., Chenu, C., Houot, S., Jolivet, R., Kätterer, T., Lutfalla, S., Macdonald, A., van Oort, F., Plante, A. F., Savignac, F., Soucémarianadin, L. N., and Barré, P.: A model based on Rock-Eval thermal analysis to quantify the size of the centennially persistent organic carbon pool in temperate soils, Biogeosciences, 15, 2835–2849, https://doi.org/10.5194/bg-15-2835-2018, 2018.

European Commission: Soils of the European Union., Joint Research Centre. Institute for Environment and Sustainability., Publications Office, LU, 2008.

Jenkinson, D. S. and Coleman, K.: Calculating the annual input of organic matter to soil from measurements of total organic carbon and radiocarbon, Eur J Soil Science, 45, 167–174, https://doi.org/10.1111/j.1365-2389.1994.tb00498.x, 1994.

Petersen, B. M., Berntsen, J., Hansen, S., and Jensen, L. S.: CN-SIM—a model for the turnover of soil organic matter. I. Long-term carbon and radiocarbon development, Soil Biology and Biochemistry, 37, 359–374, https://doi.org/10.1016/j.soilbio.2004.08.006, 2005.

Strobl, C., Malley, J., and Tutz, G.: An introduction to recursive partitioning: Rationale, application, and characteristics of classification and regression trees, bagging, and random forests., Psychological Methods, 14, 323–348, https://doi.org/10.1037/a0016973, 2009.